# Hydrodynamics of sponge pumps and evolution of the sponge body plan

Seyed Saeed Asadzadeh[1]*, Thomas Kiørboe[1], Poul Scheel Larsen[2], Sally P Leys[3], Gitai Yahel[4], Jens H Walther[2,5]

[1]National Institute of Aquatic Resources and Centre for Ocean Life, Technical University of Denmark, Lyngby, Denmark; [2]Department of Mechanical Engineering, Technical University of Denmark, Lyngby, Denmark; [3]Department of Biological Sciences, University of Alberta, CW 405 Biological Sciences Building, Edmonton, Canada; [4]The Faculty of Marine Science, Ruppin Academic Center, Michmoret, Israel; [5]Computational Science and Engineering Laboratory, Swiss Federal Institute of Technology Zürich, Zürich, Switzerland

**Abstract** Sponges are suspension feeders that filter vast amounts of water. Pumping is carried out by flagellated chambers that are connected to an inhalant and exhalant canal system. In 'leucon' sponges with relatively high-pressure resistance due to a complex and narrow canal system, pumping and filtering are only possible owing to the presence of a gasket-like structure (forming a canopy above the collar filters). Here, we combine numerical and experimental work and demonstrate how sponges that lack such sealing elements are able to efficiently pump and force the flagella-driven flow through their collar filter, thanks to the formation of a 'hydrodynamic gasket' above the collar. Our findings link the architecture of flagellated chambers to that of the canal system, and lend support to the current view that the sponge aquiferous system evolved from an open-type filtration system, and that the first metazoans were filter feeders.

*For correspondence: sesasa@aqua.dtu.dk

Competing interests: The authors declare that no competing interests exist.

## Introduction

Many aquatic suspension feeders use cilia and flagella to generate feeding flows from which they capture prey particles. Unlike ciliary arrays that drive fluid tangential to their attachment surface (*Brooks and Wallingford, 2014*; *Narematsu et al., 2015*; *Gilpin et al., 2020*), flagellated collar-cells (choanocytes) in sponges must pump water perpendicular to and through a perforated surface, working against a pressure resistance. Several choanocytes are closely packed inside chambers that are connected to inhalant and exhalant canal systems. The canal architecture and its associated pressure resistance impacts the operating condition of choanocyte chambers, and dictates morphological adaptation in such pumping units. Most sponges possess an elaborate aquiferous system (leucon) (*Reiswig, 1975a*), but some Calcarea, a small class of the Porifera, have a less elaborate aquiferous system (ascon and syconoid) (*Manuel, 2006*) *Figure 1*.

Asconoid forms are organized as a single tube in which the choanocytes form a single layer on the inner surface of the tube wall. Syconoid forms, in turn, possess many ascon-shaped cylindrical chambers branching off a central cavity (atrium). Water enters through short canals or directly through openings (ostia) into the chambers and leaves through a single outlet (apopyle) to the atrium and from there out of the osculum. In leuconoid forms, the holes in the surface of the body are connected to a complex canal system with a multitude of small chamber pump units.

Although these are often depicted as a progressive gain in complexity in the evolution of leuconoid from ascon and sycon forms, phylogenetic analyses indicate that in Calcarea both leucon and sycon type sponges most likely arose independently from ascon forms (*Manuel, 2006*), and some evidence points to the opposite process in which ascon and sycon forms may have derived from a

rigid leucon body architecture (*Dohrmann et al., 2006*). It is unknown what gave rise to the leuconoid forms of the other three sponge classes. One of the greatest modern puzzles is whether Porifera or Ctenophora branched first in the evolution of multicellular animals (*Whelan et al., 2015*; *Telford et al., 2016*; *Feuda et al., 2017*). Therefore, understanding how the sponge filtration system came about would go a long way to helping resolve whether sponges evolved from a colonial unicellular ancestor (Porifera first hypothesis) or from a tissue-grade metazoan (as in some interpretations of the Ctenophora first hypothesis).

Both choanocytes and choanoflagellates, a group of free-living unicellular and colonial flagellates, share ancestry to animals, and possess similar collar-flagellated cells, to generate adequate feeding currents. In both cells, presence of a flagellar vane, a sheet-like structure along the length of the flagellum, has been observed (*Afzelius, 1961*; *Fjerdingstad, 1961*; *Mehl and Reiswig, 1991*; *Weissenfels, 1992*; *Leadbeater, 2006*; *Mah et al., 2014*). Vanes differ from mastigonemes, which are extracellular nanometer-thick fibers that protrude perpendicular to the flagellum (*Bouck, 1971*;

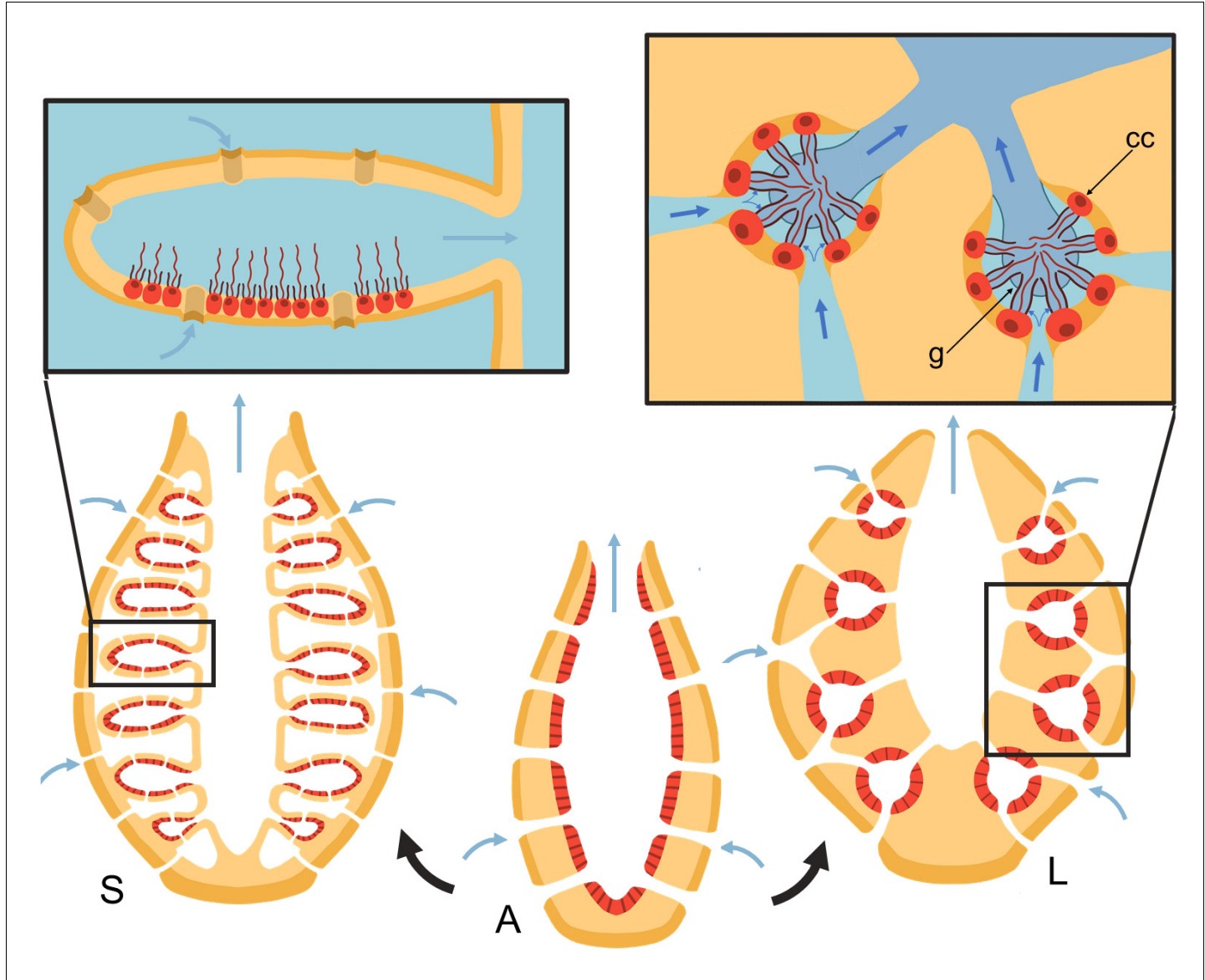

**Figure 1.** Schematic of sponge body types in ascon (A), sycon (S), and leucon (L) with evolutionary view of grades of morphological complexity in calcareous sponges (black arrows). Insets show zoom-in water path (blue arrows) into the chambers in sycons and leucons, where in the latter, water is forced to go through collar filters (black) of choanocytes (red, cc) due to the presence of a physical gasket (g).

*Fenchel, 1982*), both morphologically and functionally. Mastigonemes appear to exist and remain in the beat plane and are thought to reverse the thrust generation by the flagellum (*Holwill and Peters, 1974*). Vanes, however, are horizontal fibers of glycocalyx that extend perpendicularly from the flagellar axis and appear to augment the thrust generation (*Nielsen et al., 2017*).

Choanocytes in sponges, in addition to driving the flow through the collar, must also overcome the pressure resistance arising from the canal system. The narrow canals in leuconoid body plans add significant pressure resistance to the pump, between 2 and 50 Pa for different sponge species (*Larsen and Riisgåd, 1994*; *Leys et al., 2011*; *Ludeman et al., 2017*). Our recent study (*Asadzadeh et al., 2019a*) demonstrated that directional flow and efficient filtration at such hydrodynamic conditions is only possible because the pumping unit is sealed against back flow, by three extracellular elements, namely, a tight mesh on the distal part of the collar, a vane on the flagellum, and finally a gasket-like structure called Sollas' membrane (*Dendy, 1888*; *Sollas, 1889*), that connects the microvilli (collar elements) at their tips (*Figure 1*). These features divide the chamber into two regions of low and high pressure, and force the inflow through the collar slits at its base. In contrast, calcareous sponges have fewer canals and thus have less pressure resistance, and appear to lack the gasket and tight mesh that covers the collar (*Dendy, 1888*; *Eerkes-Medrano and Leys, 2006*).

Here, we ask how sponges with apparent absence of a gasket, and therefore an open aquiferous system, can nevertheless effectively pump and filter water for prey, and what the trade-offs and limitations in such pumping systems are. We combine modeling with experiments to study the pumping and feeding in calcareous sponges using *Sycon coactum* as a model organism. Because of the geometric complexity, rather than attempting to use singularities to model the low Reynolds flow, as done for example for flows near a hole in a plane wall (*Davis et al., 1981*), or flagella-driven flows (*Roper et al., 2013*), we use computational fluid dynamic (CFD) simulations to investigate the functionality of the pumping elements in the absence of a gasket. Specifically, we study the role of each pumping element in pumping sufficient water through the ostia and into the chamber and examine the mechanism that forces the inhalant flow to pass through the collar filter. The modeled pumping rates are compared to measured exhalant flow rates and the predicted retention efficiency of calcareous sponges is contrasted with actual retention efficiencies measured for different prey types. The limitations and trade-offs between the pumping rate and retention efficiency shed light on what may have been the first poriferan filtration system.

## Morphology of the flagellated chamber

Each flagellated chamber in *Sycon coactum* contains thousands of choanocytes that pump water through hundreds of ostia (*Leys and Eerkes-Medrano, 2006*) *Appendix 1—figure 1*. To reduce the computational costs, we consider a cylindrical axisymmetric and axially periodic chamber. The computational domain is thereby reduced to a wedge section into the cylindrical chamber that includes a central ostium surrounded by 24 choanocytes (*Figure 2*). The boundary conditions (BCs) are periodic along the long axis of the chamber (due to the periodic occurrence of ostia) and laterally symmetric on the left and right sides of the section modeled (if one is facing the long axis), due to cylindrical symmetry of the tubular chamber. Once fluid leaves the computational domain vertically into the inner cylindrical core of the choanocyte chamber, it is directed axially toward the open end of the chamber (apopyle). Pressure variation inside the core of the chamber is negligible (*Appendix 1—figure 2*), hence a uniform pressure boundary is applied on the top surface of the computational domain.

The collars are modeled as a porous structure (*Equation 6*) with a porosity that corresponds to that of a network of parallel and equally spaced cylinders (*Keller, 1964*) representing the microvilli of 0.1 µm in diameter. Although recent observations (*Appendix 1—figure 2*) have shown some mesh structures appearing locally between collars, chambers in *S. coactum* appear to lack a solid gasket. Hence we do not include a physical gasket in the CFD model. However, to study the effect of the gasket, we also consider cases where a physical gasket is included in the model. (*Appendix 2—figure 1*). Each flagellum is modeled as a thin sheet of width $W$ (resembling the vane, *Appendix 1—figure 2*) that beats in a plane and is subject to a no-slip and no-penetration boundary condition with a prescribed motion given in *Equation 3*.

We initially consider mean measured values for the dimension of the chamber elements: spacing between flagella of 5 µm, collars of diameter $D_{col} = 2.5\,\mu\mathrm{m}$, and length $L_{col} = 4.8\,\mu\mathrm{m}$ with spacing

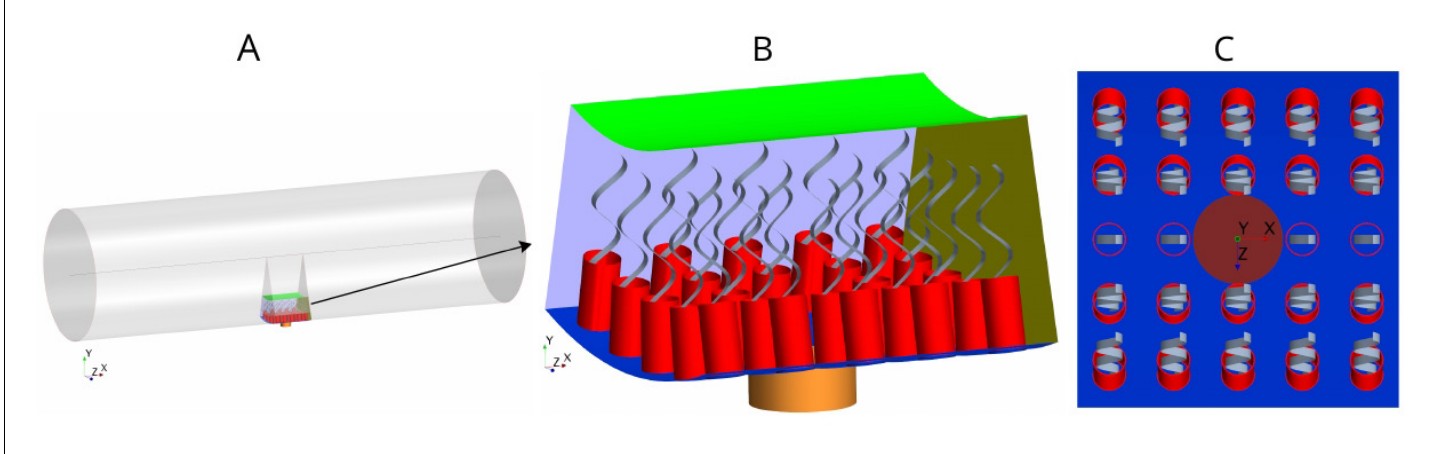

**Figure 2.** Morphology of the modeled flagellated chamber in calcareous sponges. (A) The computational domain is a wedged section into the cylindrical, ascon-like flagellated chamber (85 μm in diameter). (B) A side view magnification of the modeled section. The water is pumped into the domain through a central tubular ostium, its inlet (brown, shown in C) is subjected to a uniform pressure and its surface (orange) to no-slip boundary conditions. The domain includes 24 porous collars (red) with 24 flagella (gray) attached to the inner surface of the chamber (blue) and subjected to no-slip boundary conditions. Two sides of the domain that are perpendicular to the x axis (olive, for clarity only one side shown) are subject to periodic boundary conditions, and the other two sides that are parallel to both the x and y axis (light blue, one side shown) are subjected to symmetric boundary conditions. The water leaves the domain from the top (green, subjected to a uniform pressure boundary conditions) into the chamber core. (C) Top view showing arrangements of the inlet and nearby collars.

between microvilli of $l = 0.05\,\mu\text{m}$, and an ostium of diameter $D_{ost} = 7.0\,\mu\text{m}$ and length $L_{ost} = 3.0\,\mu\text{m}$. Flagella have a wavelength of $\lambda = 5.0\,\mu\text{m}$, an amplitude of $a = 1.0\,\mu\text{m}$ (with length scale $\delta = 1.0\,\mu\text{m}$, see *Equation 3*) and vane width of 0.7 μm, all beating in phase at a frequency of 30 Hz (base case). In subsequent experimental simulations, these dimensions are varied along with the dimensions of other elements involved in the pumping system (e.g. length of the collar, length and diameter of the ostium, porosity of the collar etc.).

## Results and discussion

### Pumping mechanism

To facilitate efficient feeding, particles carried in the inhalant flow should reach the collars where the prey particles are filtered (*Reiswig, 1971*; *Reiswig, 1975b*; *Weissenfels, 1976*; *Imsiecke, 1993*). To investigate how the flow is forced through the collar rather than bypassing it in the absence of a gasket, we examine the average velocity field in two perpendicular planes in the middle of the computational domain (*Figure 3AB*). The velocity fields reveal a large backflow in the middle of the domain above the ostium where the spacing between the flagella is large. The pressure drop after the ostium provides a suction drawing the water from the higher pressure region above the flagella to the spacing between the flagella. This flow meets the inhalant flow from the ostium nearly at the same height as that of the collar, resulting in a stagnation zone that acts as a 'hydrodynamic gasket' that forces the inhalant flow toward the collars. *Videos 1* and *2* demonstrate how formation of the hydrodynamic gasket ensures an efficient encounter of particles with the collar and prevents particles from bypassing the collars in the absence of a physical gasket.

The flow passing through the collar (filtration rate) is several times higher than the flow through the ostium (pumping rate) (*Figure 3C*), suggesting that a large proportion of the water is re-filtered inside the chamber. Increasing the vane width increases both the pumping and filtration rate, but the vane width has a larger effect on the latter (*Figure 3C*), likely due to the confinement and hydrodynamic interaction between the individual flagellum and its associated collar (*Nielsen et al., 2017*; *Asadzadeh et al., 2019b*). For instance, increasing the vane width from 0.3 to 0.7 μm increases the pumping rate by 53%, while the filtration rate is more than doubled.

To elucidate the mechanism responsible for pumping water through the ostia and to differentiate it from the mechanism that drives the water through the collars, we consider two hypothetical cases:

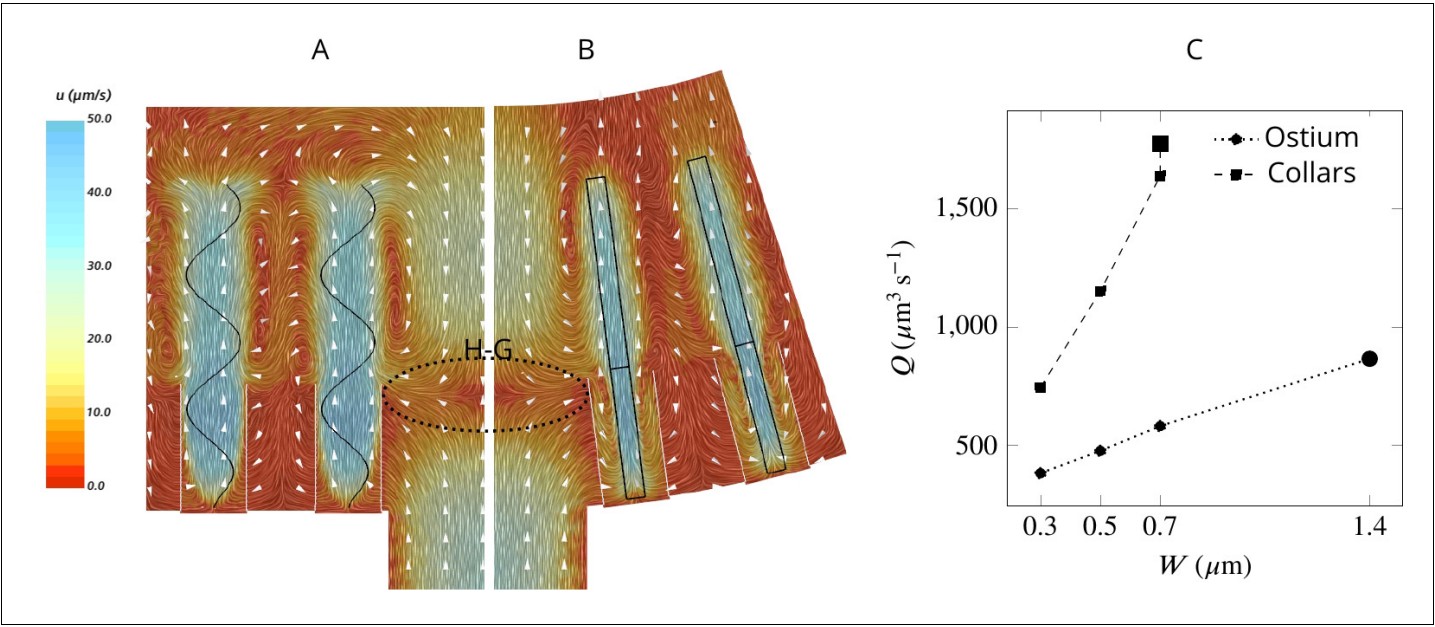

**Figure 3.** Simulated flow field inside the chamber. The averaged velocity field is plotted along the *xy*-plane (**A**) and *yz*-plane (**B**). Colors represent water velocity magnitude according to the color bar at the side; flow direction is indicated by white arrows; the flagella are indicated by black lines and the collars by gray lines. Due to the symmetry with respect to the *yz*-plane only one half of each of the planes is shown. A zone of stagnant water is formed by the backflow in the relatively large spacing between the flagella and above the ostium (H–G). This backflow serves as an effective 'hydrodynamic gasket' that forces the inflow through the collars. (**C**) Pumping rate (Q) through the ostium and filtration rate through the collars (Q) for different width (W) of the flagellar vane. The larger symbols correspond to the case where the vane width is 1.4 µm on the unconfined part of flagellum and 0.7 µm on the confined part.

(1) with flagella that beat only above the collars (*Figure 4A*) and (2) with short flagella confined to the collar height (*Figure 4B*). Flagella that are only present outside and above the collars are as efficient as complete flagella in pumping water into the chamber but are not able to drive water through the collar *Figure 4C*. In contrast, short flagella that are confined inside the collars are almost as efficient as complete flagella in driving water through the collars but are not able to pump new water through the ostium into the chamber. These results demonstrate two separate contributions from the confined and unconfined part of the flagellum of calcareous sponges. This pumping mechanism is distinctly different from that of demosponges and glass sponges, both of which possess a tissue or mucus gasket, in which the confined part of the flagellum with vanes that span the full diameter of the fine-meshed collars (*Mah et al., 2014*) is the pumping unit (*Asadzadeh et al., 2019a*).

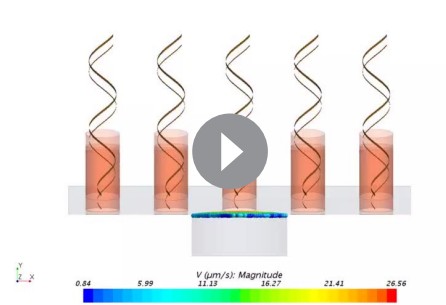

**Video 1.** Side view of passive particles entering into the chamber through the ostium and carried by the flow. Particle color denotes its velocity according to the color scale at the bottom. Particles that arrive at the collar are removed from the simulation.
https://elifesciences.org/articles/61012#video1

## Pump characteristic curve

To analyze the functionality of the pump at different hydrodynamic conditions, we consider the pump characteristic curve. In so doing, one should differentiate the basic pumping unit (pressure-generating unit) from the pressure-resistive parts of the system (the canal system). The basic pumping units are the simplest subdivision of the choanocyte chamber exposed to the same pressure difference resulting from the canal system. Hence, these units all work in parallel. For sponges lacking a physical gasket, the unit is one hole (ostium of zero length) with several

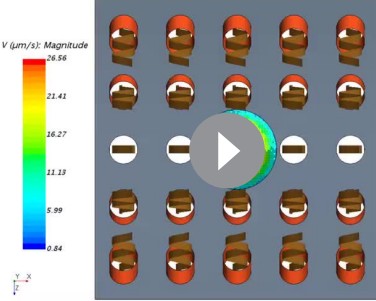

**Video 2.** Top view of passive particles entering into the chamber through the ostium and carried by the flow. Particle color denotes its velocity according to the color scale at the bottom. Particles that arrive at the collar are removed from the simulation.
https://elifesciences.org/articles/61012#video2

neighboring choanocytes (*Figure 2B*). Presence of a physical gasket, however, leads to two zones of high and low pressure inside the spherical choanocyte chambers (*Weissenfels, 1992*; *Asadzadeh et al., 2019a*), effectively exposing each individual choanocyte to the same pressure difference. Therefore, in sponges with a physical gasket, the basic pumping unit is only one choanocyte, irrespective of its proximity to the nearest ostium (*Asadzadeh et al., 2019a*).

Basic pumping units are characterized by a maximum pumping rate $Q_{max}$ (at zero pressure load), and a maximum pressure $P_{max}$ (at zero net flow). Due to the linearity of the governing equations at low Reynolds numbers, the pump characteristic of the unit is linear, hence:

$$\hat{P} = 1 - \hat{Q} \qquad (1)$$

where $\hat{P} = P/P_{max}$ and $\hat{Q} = Q/Q_{max}$ are normalized pressure and pumping rate, respectively. Assuming Poiseuille flow in the canal system, and a tubular ostium of length $L_{ost}$ and diameter $D_{ost}$, the pressure resistance (of the system) is given by *White, 2011*:

$$\hat{P} = \frac{R_{ost}}{C_{pump}} \hat{Q} \qquad (2)$$

where $R_{ost} = \frac{128}{\pi} \left( \mu L_{ost} / D_{ost}^4 \right)$ is the canal resistance, $\mu$ the dynamic water viscosity, and $C_{pump} = P_{max}/Q_{max}$ is a characteristic of the pumping unit. *Figure 5* depicts the dimensionless pump (*Equation 1*) and system (*Equation 2*) characteristic curves for different sponge pumps. Intersection of the two characteristics defines the operating condition of the pumping unit. Employing typical

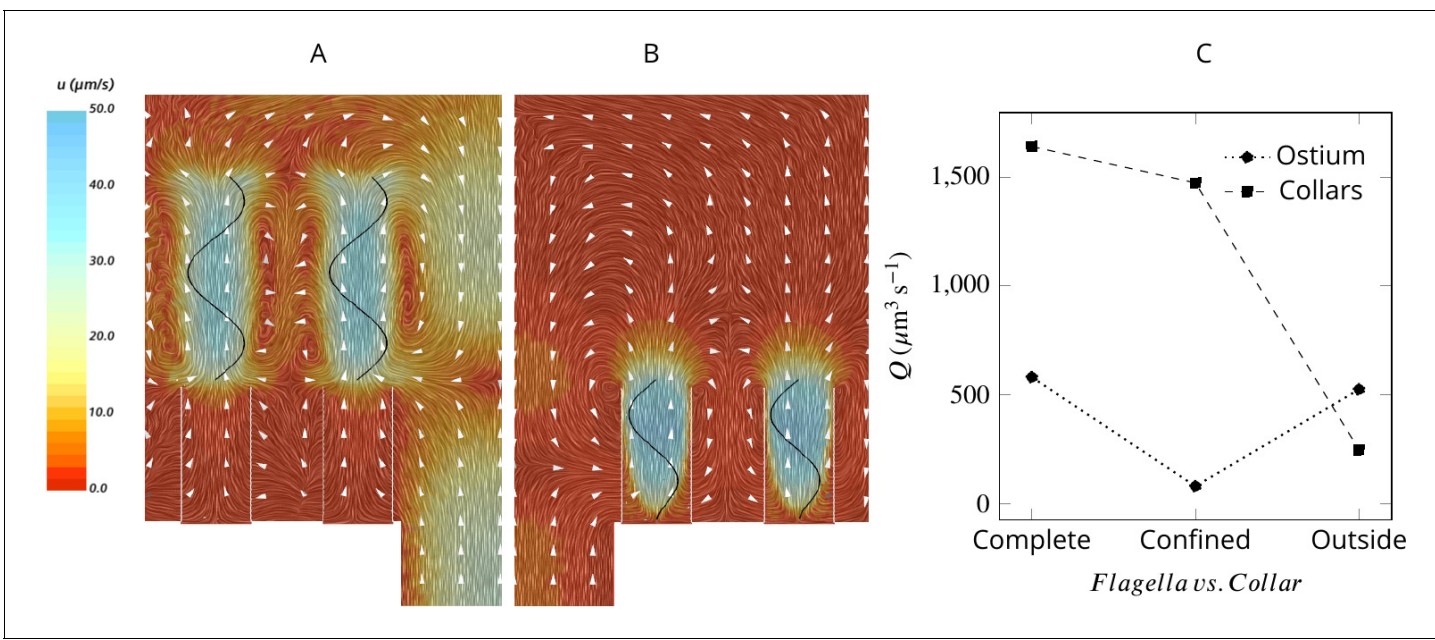

**Figure 4.** Averaged velocity field for two hypothetical cases: (**A**) only the unconfined part of the flagella exists, and (**B**) only the confined part of the flagella exists. A comparison of the pumping and filtration rates provided in (**C**) demonstrate that unconfined flagella contribute substantially to the pumping rate into the chamber, but not through the collars. On the other hand, the confined region of the flagella draws in the flow through the collars and have a minor effect on the pumping rate.

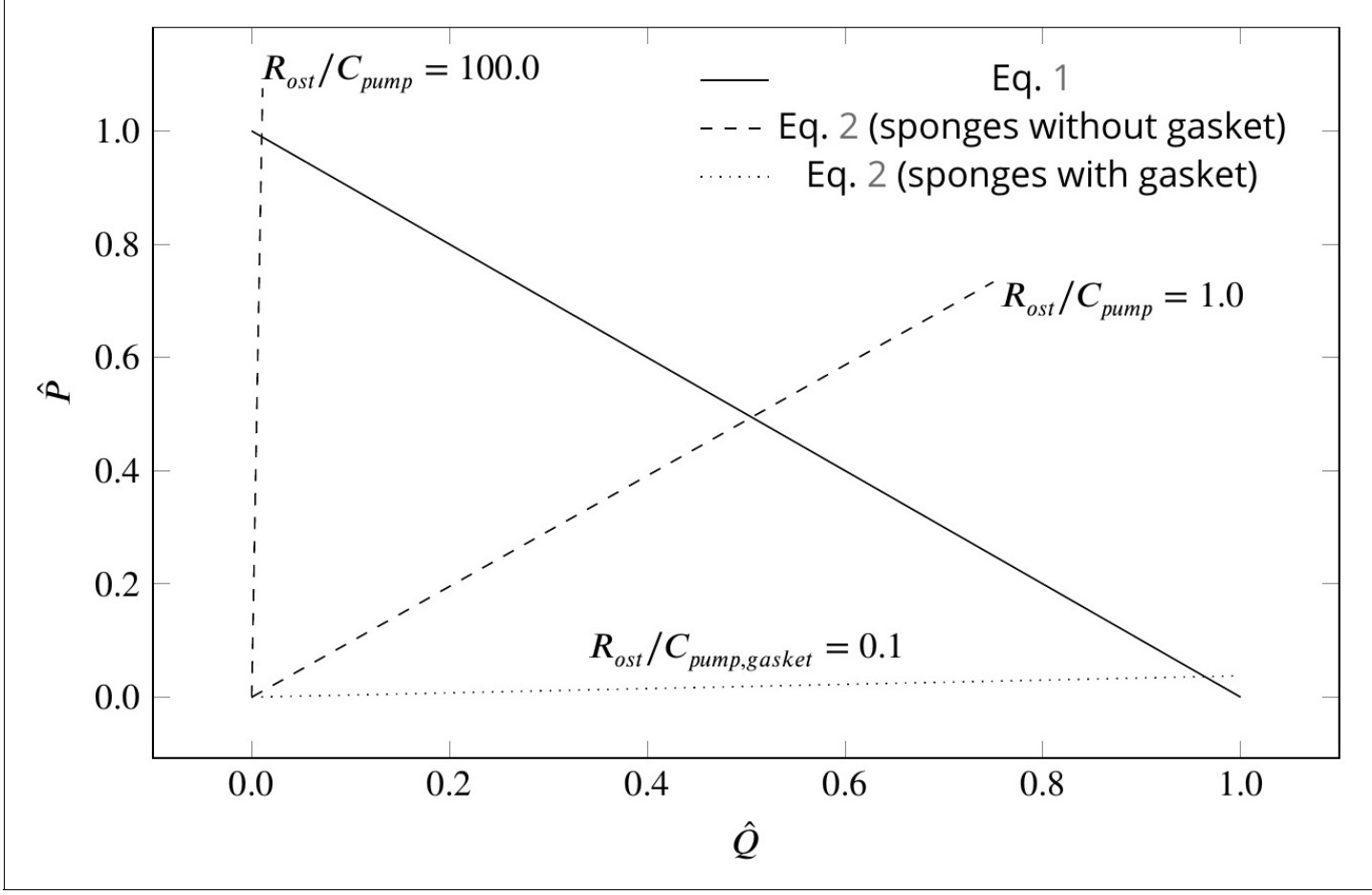

**Figure 5.** Dimensionless pump (solid line) and system (dashed and dotted lines) characteristics for different pumping units and canal systems in sponges. At their operating condition (intersection of the pump and system lines), pumping units lacking a physical gasket deliver half of their maximum pumping capacity ($R_{ost}/C_{pump} = 1.0$). Functionality of such pumps would be impaired if connected to highly resistive canal system ($R_{ost}/C_{pump} = 100$). At such hydrodynamic conditions, a modification of the pump to that of leucon type having a physical gasket ($C_{pump,gasket} \gg C_{pump}$) lowers the slope of the system curve ($R_{ost}/C_{pump,gasket} \simeq 0.1$), resulting in an efficient pumping at their operating condition.

dimensions of the ostium and $C_{pump}$ of the syconoid pumping unit (e.g. $C_{pump} = 0.058/1114 = 50\,\mu\mathrm{Pa.s}\,\mu\mathrm{m}^{-3}$ for the pumping unit in **Figure 2B**) results in $R_{ost}/C_{pump} \simeq 1$, indicating $Q \simeq 0.5 Q_{max}$ at the operating condition. The higher the resistance from the canal system, the higher the slope of the system curve, hence the lower the pumping rate. As shown in **Figure 5**, the functionality of the syconoid pumping unit would be impaired if connected to a complex and narrow canal system with two orders of magnitude higher pressure resistance (e.g. $R_{ost}/C_{pump} = 100$). For such hydrodynamic conditions, a modification of the pumping unit to that of the leucon type employing a physical gasket is inevitable. Thanks to the sealing elements, $C_{pump,gasket}$ in such pumps are much higher (typically 3 orders of magnitudes) than syconoid ones (**Asadzadeh et al., 2019a**), resulting in a much lower slope of the system curve ($R_{ost}/C_{pump,gasket} \simeq 0.1$) yielding $Q \simeq Q_{max}$ at the operating condition. Note that the actual characteristic curve in the leucon pump is non-linear, which has been ascribed to the bending of the vane at the relatively higher working pressure (**Asadzadeh et al., 2019a**).

Although sponges with a complex canal system appear to have a 'gasket-present' pumping unit, a 'gasket-absent' unit can still function efficiently if connected to a complex, yet open and less resistive canal system, or inefficiently if connected to an open canal system, but with highly resistive ostia. Therefore, there could well be sponges characterized as leuconoid types yet missing a physical gasket, or as syconoid types but with a physical gasket. The appearance, structure, and nature of the gasket would also depend on the hydrodynamics dictated by the canal system. Therefore, further

research, using better preservation and imaging techniques, is required to elucidate the exact nature of the sealant element, and to investigate the role that hydrodynamics is playing in forming the seal in different sponges.

## Hydrodynamic trade offs in the morphology of sponges

The open architecture of the syconoid pump dictates a delicate trade-off between the pumping rate and the retention efficiency. Although a wider and shorter ostium provides less resistance to flow hence higher pumping rate, the reduced pressure loss comes with a cost of a weaker backflow in the core of the region and thus a weaker hydrodynamic gasket located further into the chamber and above the collar tips. Under such circumstances, a larger fraction of the flow bypasses the collars and the retention efficiency drops. Lower retention efficiency of the calcareous sponge *Pericharax heteroraphis* as compared to different species of demposponges has been previously ascribed to its wider ostia and large chambers (*Wilkinson, 1978*). In contrast, a narrow and long ostium with higher resistance to flow results in a reduced pumping rate but an increased retention efficiency as the hydrodynamic gasket is located below the tip of the collars and the stronger backflow is more efficient in preventing flow from bypassing the collar filter (*Appendix 1—figure 4*).

The dependency of the pumping rate into the chamber on the length of the collar suggests another trade-off: choanocytes with shorter collars and a longer unconfined part of the flagellum can pump more water into the chamber, but this comes at the cost of more flow bypassing the collar, and vice versa. As a result, regions of the chamber having shorter collars will be less efficient in filtering the inhalant flow. This effect may explain the presence of pseudopodial extensions extending from the cell surface in these regions (*Leys and Eerkes-Medrano, 2006*). The pseudopodia, which are on average twice as long as the collars, have been observed to reach beyond the collar and seem to be involved in particle capture (*Leys and Eerkes-Medrano, 2006*), potentially compensating for the decreased filtration efficiency of the collars.

On individual collars, the spacing between adjacent microvilli appears wider at the base but becomes smaller toward the tip of the collar where the microvilli tips are occasionally fused together (*Leys and Eerkes-Medrano, 2006*). To study the effect of non-uniform porosity along the collars on the performance of the sponge pump, we consider cases where the spacing between microvilli decreases from the base to the tip of the collar. We find that more widely spaced (higher porosity) microvilli on collars enhance the filtration rate with a minimal effect on the pumping rate into the chamber (compare the plots in *Appendix 1—figure 5*). This result further highlights the fact that the pumping rate into the chamber is nearly independent of the confined part of the flagellum, while the filtration rate is greatly affected by the local collar-flagella parameters.

## Flagella kinematics

Thus far, we have considered a synchronous beat and constant amplitude. We now consider different scenarios, that is, phase shift among the flagella, flagella beating in different planes (*Appendix 1—figure 6*), different frequencies, and different wavelength (*Appendix 1—figure 7* and *8*). The results show that beat synchronization among choanocytes is not necessary for the functionality of the pump. Despite improved efficiency of pumping in ciliary arrays by synchronization (*Niedermayer et al., 2008*; *Golestanian et al., 2011*; *Elgeti and Gompper, 2013*), asynchronized flagella in sponges continue to pump efficiently into the chamber (discussed in details in the SI). Lack of synchronization has been reported both within colonies of the choanoflagellate *Salpingoeca rosetta* (*Roper et al., 2013*; *Kirkegaard et al., 2016*), and in choanocyte chambers of the freshwater sponge *Spongilla lacustris* (*Mah et al., 2014*).

The amplitude of the flagella waveform is limited inside the collars, but it can increase outside, a behavior observed in choanocytes of the *S. lacustris* (*Mah et al., 2014*). To study the effect of increased amplitude, we modify the beat form in *Equation 3* by setting $a = 5\,\mu\mathrm{m}$ and $\delta = 22\,\mu\mathrm{m}$, which results in the increased amplitude of the waveform to a maximum of 2 μm at the tip of the flagella. This modulation in the beat form improves pumping rate by 51% (to $Q_{ost} = 874\,\mu\mathrm{m}^3\,\mathrm{s}^{-1}$) while keeping the stagnation area at the same height relative to the base (*Figure 3A*), but it is only 37% more energetically demanding, suggesting that such a modification is beneficial to the sponge.

To compare the CFD predictions with observations, we estimate the volume flow rate per ostium in *Sycon coactum* by dividing the experimentally measured exhalant flux rate by the estimated total

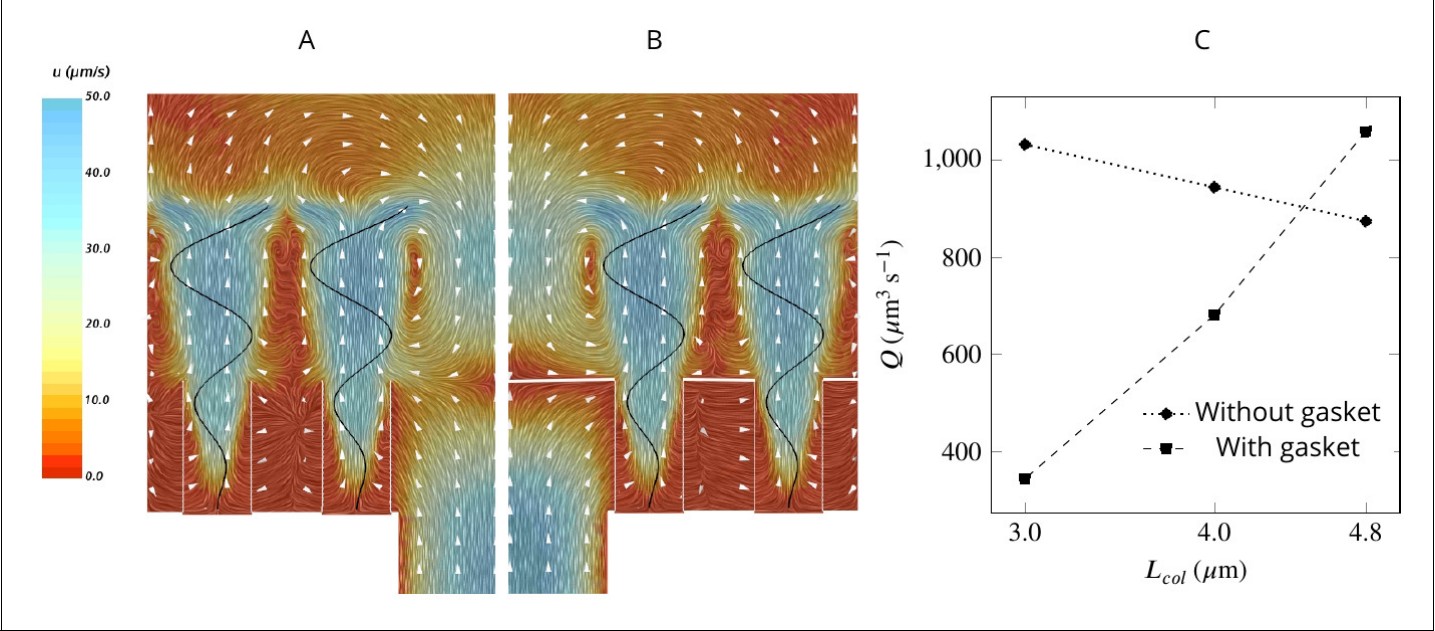

**Figure 6.** A comparison of mean velocity fields and the performance of (A) a hydrodynamic, and (B) a physical gasket (both with an increased amplitude of the flagella beat waveform and collar length of 4.8 µm). (C) The volume flow rate through the ostium for different collar lengths. Inclusion of the physical gasket does not significantly affect the flow pattern inside the chamber. In sponges with relatively short collars (calcareous sponges), choanocytes pump more through the ostium the shorter the collar but increasing less through the collars (C).

number of ostia. This results in an estimated flow rate of $Q_{ost,exp} = 1151\,\mu\mathrm{m}^3\,\mathrm{s}^{-1}$ per ostium, and $Q_{ch,exp} = 48\,\mu\mathrm{m}^3\,\mathrm{s}^{-1}$ per choanocyte (given 24 choanocytes). CFD results show that ascon and sycon type sponges with typical values and dimensions of the morphological elements, are able to pump more than $Q_{ost} \sim 1000\,\mu\mathrm{m}^3\,\mathrm{s}^{-1}$ through an ostium without any requirement for a physical gasket (e.g. for the case with increased amplitude and a vane width of 1.4 µm on the unconfined part of the flagellum, $Q_{ost} = 1295\,\mu\mathrm{m}^3\,\mathrm{s}^{-1}$). Furthermore, both experimental and CFD estimates of flow rate per choanocyte ($Q_{ch,exp}$) are comparable to published estimates for leucon sponges that range from 17 to $236\,\mu\mathrm{m}^3\,\mathrm{s}^{-1}$ for different species of demosponges and glass sponges (*Larsen and Riisgård, 1994*; *Leys et al., 2011*; *Ludeman et al., 2017*), suggesting similar pumping capacity despite different pumping mechanism.

## Effect of a gasket

To study the effect of a physical gasket on the flow and pumping rate, we model this structure as an impermeable baffle and incorporate it into the computational domain (*Appendix 2—figure 1*). The inclusion of a physical gasket improves the pumping rate by 21% if the collar is long (4.8 µm), but it does not alter the pattern of the averaged flow much (*Figure 6AB*). However, with shorter collars, a physical gasket decreases the pumping rate, and the more so the shorter the collar (*Figure 6C*). This phenomenon may explain the shorter collars in calcareous sponges compared to those in demosponges (*Leys and Eerkes-Medrano, 2006*) .

## Retention efficiency

Simulations of passive prey particles (*Videos 1* and *2*) show that the prey retention efficiency is potentially 100%. However, these simulations assume that all particles encountering the filter are retained. In reality, however, the retention efficiency can vary for different prey types, especially in the absence of a gasket sealing off the collar filter area from the rest of the chamber. We studied experimentally the retention efficiency in *Sycon coactum* of three different prey types: small microalgae (Euk < 20 µm), and non-photosynthetic bacteria with high (HNA) and low (LNA) nucleic acid content. While Euk and HNA were retained with near 100% efficiency, the efficiency of retention for the LNA prey particles was considerably lower (60 ± 6%) (*Figure 7*). LNA is largely associated with

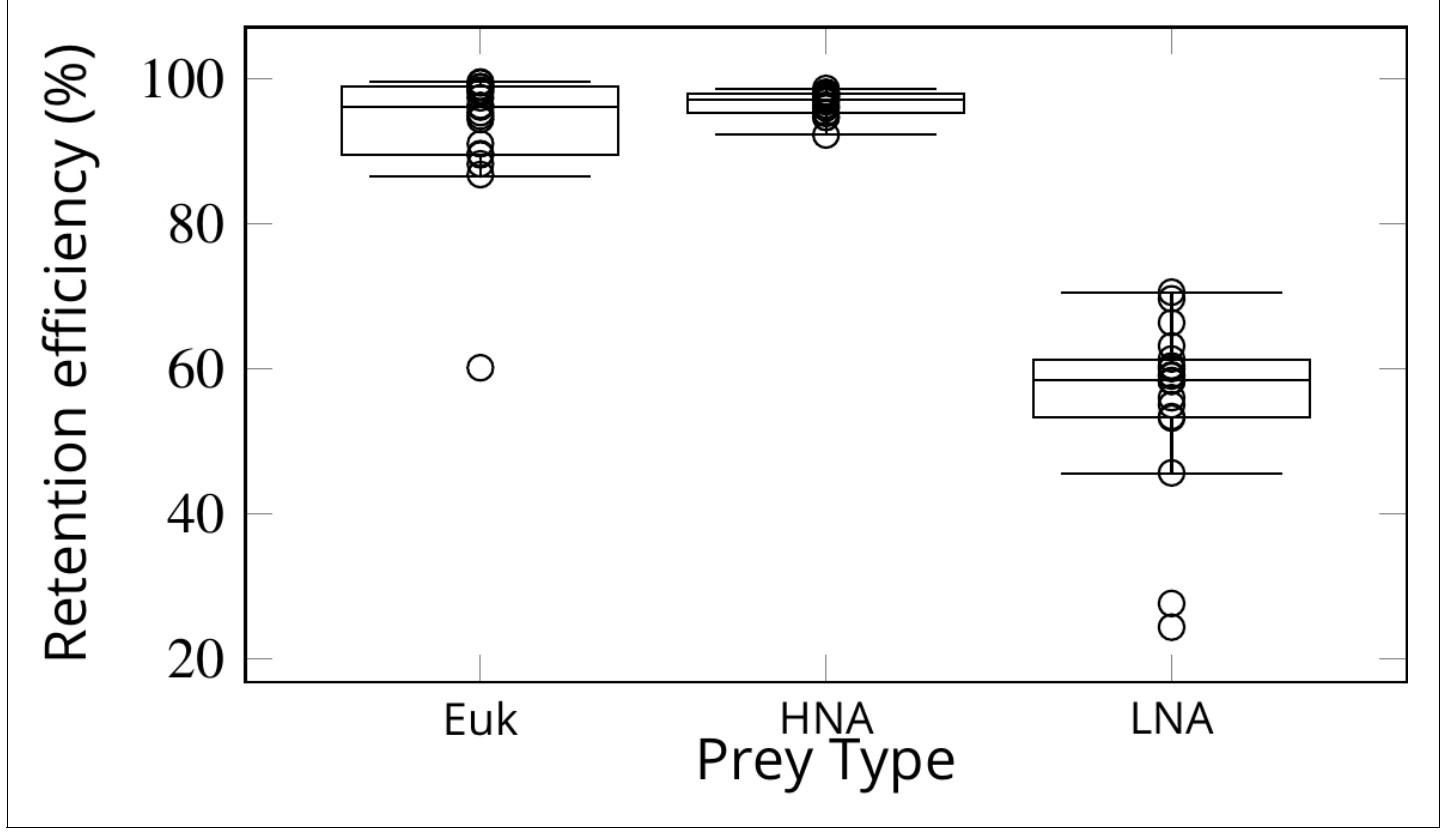

**Figure 7.** Retention efficiency (%) of different prey types counted by flow cytometry in the water inhaled and exhaled by *Sycon coactum*. Euk are small nano and pico eukaryotic algae; HNA and LNA are non-photosynthetic bacteria with high and low nucleic acid content, respectively. Centre lines in each box show medians; box limits indicate the 25th and 75th percentiles; whiskers represent local minima and maxima, that is, they extend to data points that are less than 1.5 × IQR away from the 25th and 75th percentiles (*IQR* is the interquartile range), outliers are represented by dots.

The online version of this article includes the following source data for figure 7:

**Source data 1.** Sycon feeding and filtration.

SAR11, the smallest and most abundant bacterial clade in the ocean (*Mary et al., 2006*). These bacteria may slip through the mucus filter of tunicates (*Dadon-Pilosof et al., 2017*; *Dadon-Pilosof et al., 2019*) and evade filtration in many sponges (*Ribes et al., 2012*).

## Evolutionary implications

Our modeling and experimental analyses indicate that the morphology of different sponge body plans results in different hydrodynamics with associated trade-offs in the sponge pump in terms of volume filtered and retention efficiency. Although the presence of sealing elements is crucial for the functionality of the high-pressure leucon sponge pump, the open architecture of the ascon and sycon type pumps is as efficient at particle capture without these structures. Taken in this light, our findings have implications for interpreting the first poriferan body plans. Neoproterozoic oceans were food and oxygen poor, with smaller bodied animals (*Sperling and Stockey, 2018*). In a Neoproterozoic ocean, competition with colonial flagellates could have favored a filter-feeder that captured larger particles without clogging, as is the case in ascon and sycon forms. With the Cambrian explosion came additional food sources providing the energy to enable specialization of the pump and filter to capture food in a range of habitats. Our findings demonstrate that the presence of sealing elements around the sponge collar is directly related to the hydrodynamics of operating conditions associated with the body architecture, and comes with delicate trade-offs between the sponge pumping rate and retention of particles. These analyses support the view (*Manuel et al., 2003*; *Cavalier-Smith, 2017*; *Nielsen, 2019*) that the sponge aquiferous system evolved from an open-type ascon-like filtration system, and the idea that the first metazoans were filter feeders.

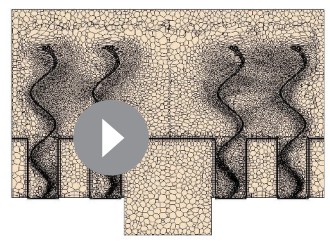

**Video 3.** The moving mesh (viewed in the *xy*-plane) inside the computational domain.
https://elifesciences.org/articles/61012#video3

## Materials and methods

### Numerical simulations

We use computational fluid dynamics (CFD) to solve the governing Navier-Stokes equations of the flow inside the computational domain (*Figure 2*). A finite volume method is used to discretize and solve the equations on a discrete representation of the computational domain consisting of polyhedral cells by applying the commercial CFD code STAR-CCM+ (14.04.013-R8). We use mesh morphing along with the overset method to move the computational mesh (Appendix 2, Section 2). The morphing motion redistributes mesh vertices in response to the movement of the flagellum. *Video 3* shows the moving mesh (viewed in the *xy*-plane), inside the computational domain. To ensure that the solutions are independent of the computational mesh sizes, the simulations (for the base case) have been repeated using four coarser and one finer meshes (*Appendix 2—figure 5*). To demonstrate the independence of the CFD results from the global boundary conditions (BC) of the flagellate chambers (i.e. the chamber being closed at one end and open at the other end (apopyle), one simulation was conducted in an extended slice of a cylindrical domain including five ostia, closed (no-slip BC) at one end and open (uniform pressure BC) at the other end (*Appendix 2—figures 3* and *4*)).

### Measurement of the pumping rate and retention efficiency in live sponges

Ten specimens of *Sycon coactum* were retrieved by scuba divers and transported at controlled temperatures to the laboratory at the Bamfield Marine Sciences Center, Bamfield, BC, Canada. Specimens were cleaned of macro-epibionts and left in large tanks with a flow-through system until the experiments were performed.

The volume flow rate through the oscula was calculated as the product of the ex-current jet speed and osculum area assuming a plug flow profile across the oscula (Dye Speed, DS, methods see *Morganti et al., 2017* and references therein). Briefly, the excurrent jet speed was measured by releasing small amounts of filtered (0.2 μm) seawater mixed with sodium fluorescein dye next to the sponge ostia and videotaping the dye front in the excurrent jet along with a known scale. This procedure was repeated 5–10 times per sponge, and the DS was measured by recording the time it took the dye front to travel a short distance (10–20 mm). The osculum area was measured from photographs of each osculum using ImageJ (*Schindelin et al., 2012*).

To measure the retention efficiency of the sponges, we used a direct comparison of prey cell concentration in the water inhaled and exhaled by the sponge as described by *Yahel et al., 2006*. For the In-Ex method experiments, samples were processed in pairs (inhaled and exhaled samples). All samples were spiked with 1.0 μm Polysciences Inc Fluoresbrite yellow-green beads (Cat # 17154) from a stock solution that had been pre-calibrated with Becton Dickinson Trucount Control beads (Cat # 340335). Instrument flow rate was determined using bead counts. For the current analysis, we excluded the picocyanobacteria as their numbers were low, and identification was uncertain. Retention efficiency (%) was calculated as: $100\,(C_{in} - C_{ex})/C_{in}$, where $C_{in}$ and $C_{ex}$ are the cell concentration in the inhaled and exhaled water, respectively.

### Acknowledgements

We acknowledge support from the Danish Council for independent Research (7014-00033B) to TK, from the Villum Foundation for JHW. through research grant no. 9278 and from NSERC Discovery Grant 2016–05446 to SPL. The Centre for Ocean Life is supported by the Villum Foundation.

## Additional information

### Funding

| Funder | Grant reference number | Author |
|---|---|---|
| Villum Fonden | 9278 | Seyed Saeed Asadzadeh<br>Poul Scheel Larsen<br>Jens H Walther |
| NSERC | 2016-05446 | Sally P Leys |
| Villum Fonden | | Seyed Saeed Asadzadeh |
| Danish Council for Independent Research Natural Sciences | 7014-00033B | Thomas Kiørboe |

The funders had no role in study design, data collection and interpretation, or the decision to submit the work for publication.

### Author contributions

Seyed Saeed Asadzadeh, Conceptualization, Software, Validation, Methodology, Writing - original draft; Thomas Kiørboe, Conceptualization, Resources, Supervision, Funding acquisition, Methodology, Writing - review and editing; Poul Scheel Larsen, Conceptualization, Formal analysis, Visualization, Methodology, Writing - review and editing; Sally P Leys, Conceptualization, Resources, Data curation, Formal analysis, Investigation, Methodology, Writing - review and editing; Gitai Yahel, Resources, Data curation, Formal analysis, Investigation, Visualization, Methodology, Writing - review and editing; Jens H Walther, Conceptualization, Resources, Software, Formal analysis, Supervision, Funding acquisition, Methodology, Writing - review and editing

### Author ORCIDs

Seyed Saeed Asadzadeh (iD) https://orcid.org/0000-0002-6360-8924
Thomas Kiørboe (iD) https://orcid.org/0000-0002-3265-336X
Poul Scheel Larsen (iD) http://orcid.org/0000-0002-7155-5965
Sally P Leys (iD) http://orcid.org/0000-0001-9268-2181
Gitai Yahel (iD) http://orcid.org/0000-0003-2306-355X

### Decision letter and Author response

Decision letter https://doi.org/10.7554/eLife.61012.sa1
Author response https://doi.org/10.7554/eLife.61012.sa2

## Additional files

### Supplementary files

• Transparent reporting form

### Data availability

All data generated or analysed during this study are included in the manuscript and supporting files.

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

## Appendix 1

### The calcareous sponge *Sycon coactum*

The general anatomy of *S. coactum* is illustrated in *Appendix 1—figure 1*. The sponge forms a single tube, 8 cm long, with a wall thickness of ~1 mm and an inner diameter of 8 mm. A 1 mm$^2$ piece of the wall was estimated to contain 36 chambers. Each chamber, in turn, was estimated to posses ~283 ostia, each surrounded by 20–25 choanocytes. An excurrent jet velocity of 7.5 mms$^{-1}$ was measured by dye visualization of the jet from an osculum with a diameter of 2 mm (*Appendix 1—video 1*).

|   A   |   B   |   C   |
|:-----:|:-----:|:-----:|

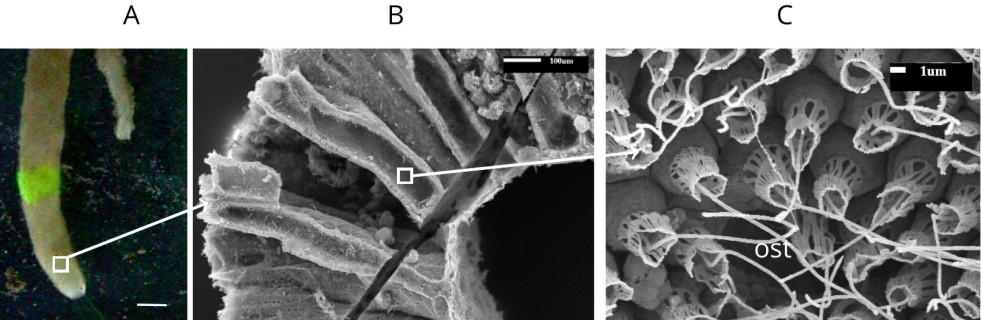

**Appendix 1—figure 1.** Calcareous sponge *Sycon coactum*. (**A**) In natural habitat (scale bar: 1 cm). (**B**) A fracture across the body of the sponge showing relatively large cylindrically shaped chambers. (**C**) An ostium (ost) with the surrounding choanocytes in the chamber. Note that although flagella here appear as smooth, they do have a vane as shown in *Appendix 1—figure 2*.

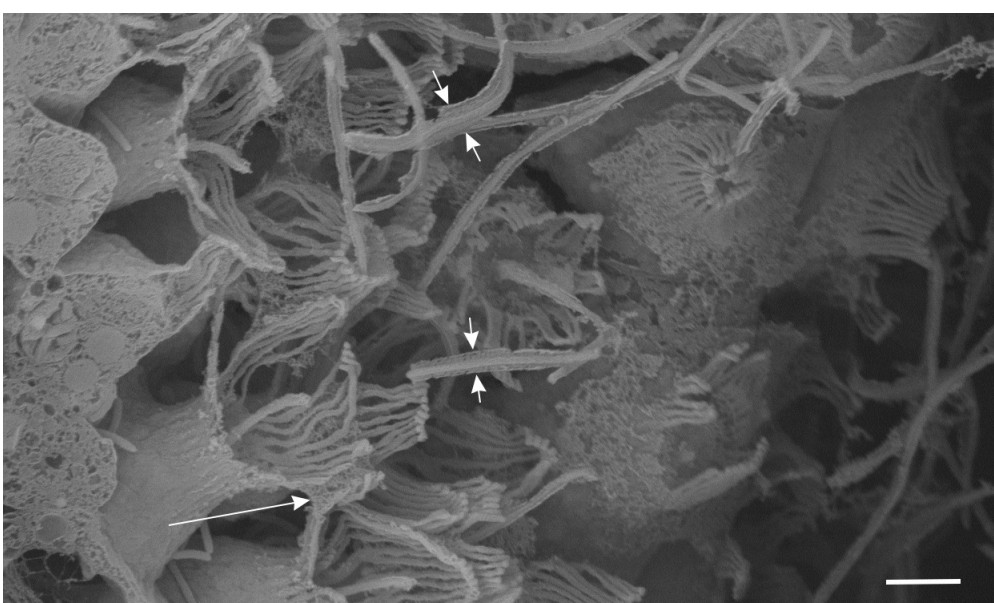

**Appendix 1—figure 2.** Observation of vane in the calcareous sponge *Sycon coactum* (scale bar: 2 μm). The flagellar vane appears as two symmetrical wing-like projection (short arrows). Some mesh structure (long arrow) appears locally between collars, attached not to the collar tips, but rather lying further down the collar and attaching to the glycocalyx mesh that occurs between collar microvilli; it possibly functions to guide flow to the collar microvilli filters.

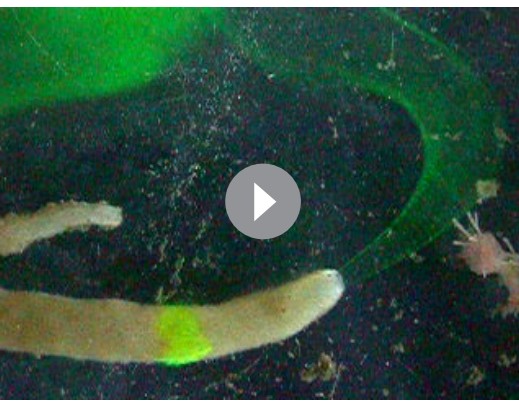

**Appendix 1—video 1.** Dye visualization of the jet from an osculum with a diameter of 2 mm.
https://elifesciences.org/articles/61012#A1video1

## Flagella beat model

The waveform of each individual flagellum is modeled as:

$$d(Y,t) = a\left(1 - e^{-(Y-Y_b)/\delta}\right)\sin\left(\frac{2\pi}{\lambda}[(Y-Y_b) - Vt]\right) \tag{3}$$

where $d$ is the local lateral displacement of the each flagellum, $a$ is the amplitude modulation of the waveform, $\delta$ is the characteristic length scale of the amplitude, $Y$ is the coordinate along the centerline axis of the flagellum, $Y_b$ is the coordinate of the flagellum at its base on the cell surface, $V = \lambda f$ is the wave speed, where $f$ is the frequency, and $\lambda$ the wavelength, and $t$ is time. Note that in the model of *Equation 3*, the flagellum would be extensible and its length varies slightly during the beat cycle (ca. 2%), resulting in marginally wrong velocities on the flagellum. However, implementing *Equation 3* in the numerical scheme is computationally cheaper than preserving the arclength. Re-running the simulation for the base case while the arclength is preserved shows that the flow field in the domain (*Appendix 1—figure 3*) is insignificantly affected with less than 4% difference in the pumping rate.

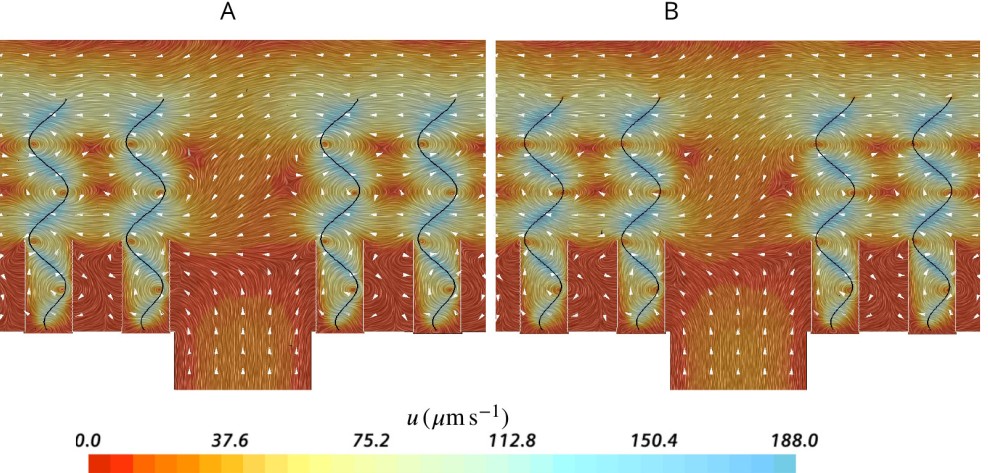

**Appendix 1—figure 3.** Snapshots of velolicty field for cases where the flagellum is extensible (**A**) and inextensible (**B**).

For the flagellate pumps, the Reynolds number ($\mathrm{Re} = \rho V L / \mu$), the ratio of inertia to viscous forces, is small ($2.9 \times 10^{-3}$), employing water density $\rho = 1022\,\mathrm{kg\,m^{-3}}$ and dynamic viscosity $\mu = 0.001\,\mathrm{Pa\,s}$, flagella length of $L = 15.5\,\mu\mathrm{m}$, flagella wave speed of $V = \lambda f = 150\,\mu\mathrm{m\,s^{-1}}$. As a result, the unsteady inertia terms in the governing equations can be neglected and flow is quasi-steady and symmetrical in respect to the flagellum beat. Therefore, to obtain the average velocity and pressure field, the simulation was conducted only for half of a beat cycles. For the particle filtration, however, the simulation is conducted for a 145 beat cycles in order to track the particles inside the domain.

The mechanical power expenditure by the flagellum is given by:

$$\mathcal{P} = \iint_{S_{fl}} \mathbf{u} \cdot (\sigma \cdot \mathbf{n})\,\mathrm{d}S \qquad (4)$$

where $\sigma = -p\mathbf{I} + \mu(\nabla\mathbf{u} + (\nabla\mathbf{u})^T)$ denotes the stress tensor, $\mathbf{n}$ is the unit normal vector on the surface $S$ pointing into the fluid, and $S_{fl}$ is the surface area of the flagella.

## Simulations of particle filtration

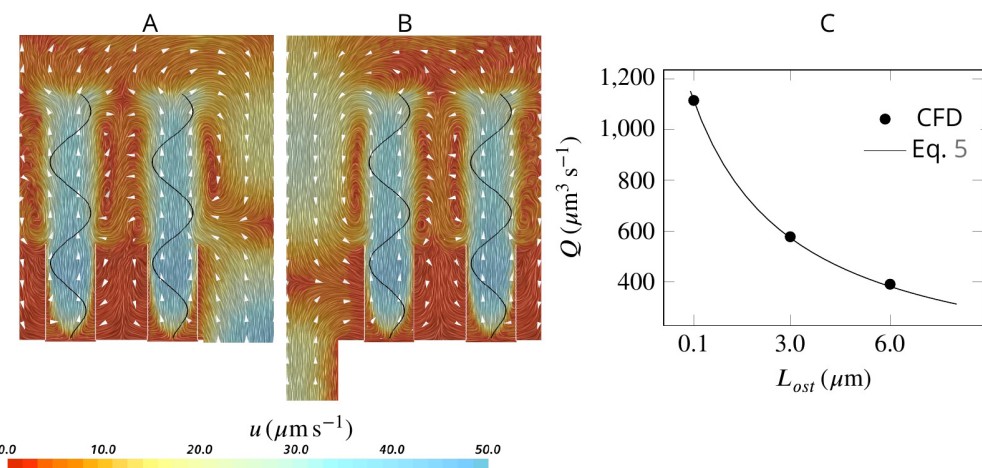

**Appendix 1—figure 4.** Effect of ostia length and diameter on the pumping rate and the backflow in the core of the region, and thus the position of the stagnation area. A very short and wide ostium (**A**) increases the pumping rate, but at the cost of a high level of flow bypassing the collar. A longer and narrower ostium (**B**), as observed in sponges, reduces the pumping rate but lowers the height of the stagnation area. (**C**) Validity of *Equation 5* vs CFD results for three different length of the ostium.

*Videos 1* and *2* illustrate passive particles entering into the chamber through the ostium and carried by the flow. Particle color denotes its velocity according to the color scale at the bottom. Particles that arrive at the collar are removed from the simulation. Note that while most of the particles are carried almost directly toward the collars, some are carried upward, above the collars, and toward the tip of the flagella, but almost all the particles are eventually returned by the back eddies and are captured. Simulation duration is 145 beat cycles. Side view (*Video 1*). Top view (*Video 2*).

## Effect of ostium length and diameter on the pumping

Assuming a fully developed flow in a tubular ostium of length $L_{ost}$ and diameter $D_{ost}$, the pressure resistance is given by *White, 2011*: $P_{ost} = \frac{128}{\pi}(\mu L_{ost}/D_{ost}^4)Q_{pump}$, where $\mu$ is the dynamic water viscosity (here $\mu = 0.001\,\mathrm{Pa\,s}$, corresponding to seawater at 20°C with salinity of 30 PSU). Note that pressure resistance due to the apopyle is 2 orders of magnitude smaller than that of the ostium (hence negligible), given the averaged dimensions of the apopyle $D_{apop} = 56\,\mu\mathrm{m}$, $L_{apop} = 17\,\mu\mathrm{m}$ and of the ostium $D_{ost} = 5\,\mu\mathrm{m}$, $L_{ost} = 5\,\mu\mathrm{m}$, and 283 ostia per apopyle. Due to the linearity of the governing equations at low Reynolds numbers, the pump characteristic of the unit is linear, hence:

$Q_{pump} = Q_{max} - Q_{max}(P_{ost}/P_{max})$, where $Q_{max}$ and $P_{max}$ are the maximum flow rate (at zero length ostia) and pressure delivered by the pump, respectively. Combining these two equations, the pumping rate of the unit as a function of the ostium dimension is:

$$Q_{pump}(L_{ost}, D_{ost}) = \frac{Q_{max}}{1 + 128\mu L_{ost} Q_{max}/(\pi D_{ost}^4 P_{max})} \tag{5}$$

Having $Q_{max}$ and $P_{max}$, *Equation 5* accurately predicts the pumping rate for a given length of the ostium as tested against CFD results *Appendix 1—figure 4C*.

## Porosity of the collar filter

We model the porosity of a collar filter composed of several microvilli of radius $b$ as

$$v_n = \kappa \frac{b}{\mu} P_c \tag{6}$$

where $v_n$ is the velocity normal to the filter subject to pressure drop, $P_c$, $\kappa$ is the dimensionless porosity of a network of parallel and equidistantly spaced cylinders (*Keller, 1964*), and $\mu$ is the viscosity of the water.

For the case of non-uniform porosity along the length of the collars (*Appendix 1—figure 5*), the dimensionless porosity $\kappa$ of the collar is modeled as $\kappa = \frac{1}{2}\kappa_b(1 + \cos(\pi y_c/L_c))$, where $\kappa_b$ is the dimensionless porosity at the base corresponding to the maximum spacing ($l_b$) between adjacent microvilli, $y_c$ the position along the collar with respect to the base, and $L_c$ the length of the collar.

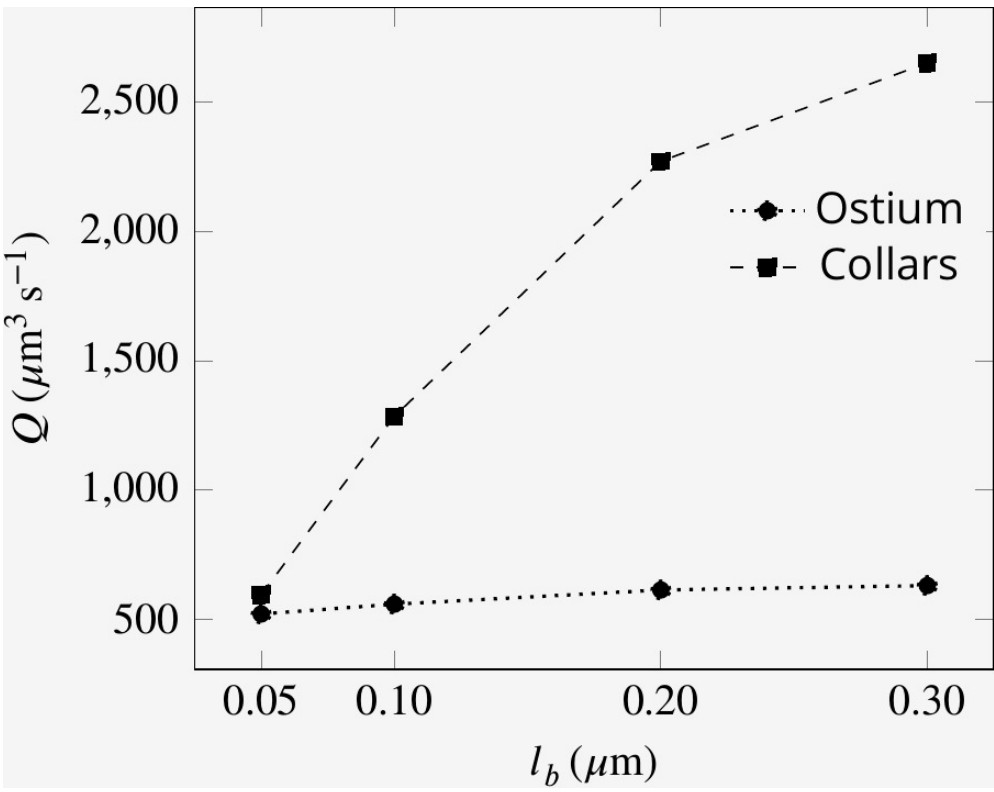

**Appendix 1—figure 5.** Effect of collar elements spacing ($l_b$) on pumping rate ($Q$) and volume flow rate ($Q$) through the collars for different porosity levels of the collars. $l_b$ is the maximum spacing between adjacent microvilli and the porosity is decreases to zero at the tip of the collars (see text).

## Asynchronization among flagella

Choanocytes are closely packed in the chambers with spacing between neighboring flagella of ~5–10 μm. At such relatively short distances, synchronization of flagella beat might be expected (**Brumley et al., 2014**), although lack of synchronization has been reported both within a colony of the choanoflagellate *Salpingoeca rosetta* (**Kirkegaard et al., 2016**; **Roper et al., 2013**), and in choanocyte chambers of the sponge *Spongilla lacustris* (**Mah et al., 2014**). To investigate the potential effect of synchronized and asynchronized flagella beat on the mechanical energy produced by the flagella, we consider three different scenarios where: (1) neighboring flagella beat slightly out of phase (phase shift angle between neighboring flagella of $\phi_f = 4\pi/25$), (2) neighboring flagella are completely out of phase ($\phi_f = \pi$), and (3) neighboring flagella beat in different (perpendicular) planes.

The snapshots of flow fields (**Appendix 1—figure 6AB**) are highly affected by the position of the neighboring flagella with respect to each other, especially near and between the flagella. However, the pumping rate and flow through the collar filter - lower and upper data, respectively, in C - are insignificantly affected by asyncronization among flagella (**Appendix 1—figure 6C**).

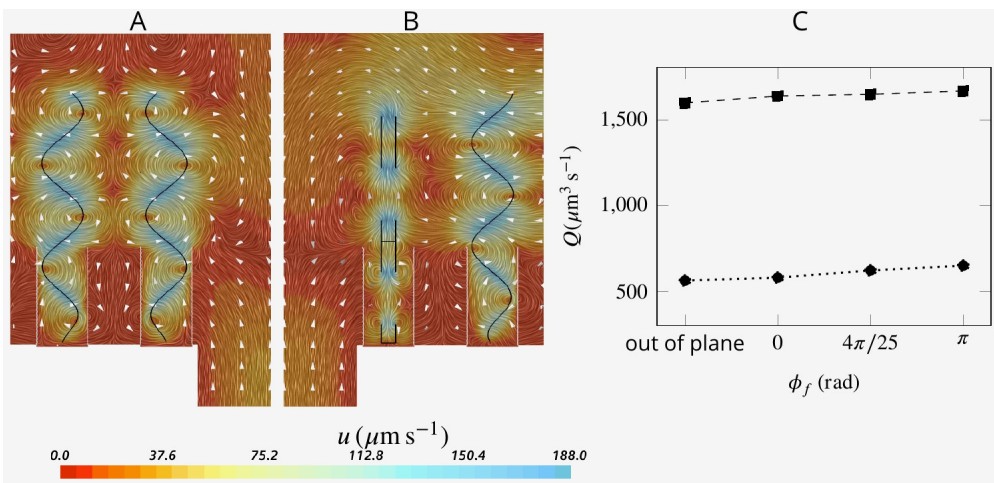

**Appendix 1—figure 6.** Snapshot of velocity fields when neighboring flagella beat completely out of phase (**A**, $\phi_f = 0$) and in perpendicular planes (**B**). Although snapshots of velocity differ for different scenarios, the pumping rate and flow through the collar filter are insignificantly affected by asyncronization among flagella (**C**).

Multi-cilia and ciliary arrays beat in phase (**Wan, 2018**) or slightly out of phase that is, metachronal waves (**Niedermayer et al., 2008**; **Golestanian et al., 2011**; **Elgeti and Gompper, 2013**) to produce strong flows tangential to the surfaces to which they are attached. When the cilia beating is completely out of phase, the tangential forces of neighboring cilia cancel each other and no flow is created. In contrast, the array of flagella that lines the inner walls of sponge chambers produces a flow that is moving perpendicular to the attachment surface (and through it). In addition, some flagella are located in proximity to the water sources (ostium) while others are more distanced. To investigate the relationships between flagella location, their synchronization, and the mechanical power produced we compare the base case in which all flagella are beating in phase to two cases where the beat of the flagella next to the ostium (eight proximal flagella) is not synchronized to that of the 16 distal flagella. In the first case, the proximal flagella are beating faster ($f_p = 60\,\mathrm{Hz}$) than the distal flagella ($f_d = 30$) and in the second case, the proximal flagella beat slower ($f_d = 30\,\mathrm{Hz}$) and the distal flagella beat faster ($f_d = 60\,\mathrm{Hz}$**Appendix 1—figure 7AB**). Results show that pumping rate is not impaired by such asynchronous flagella beat. Moreover, pumping rate of the first case (where proximal flagella pump faster (**Appendix 1—figure 7A**)) is 22% higher than that of the second case where the distal flagella beat faster ($Q_1 = 984\,\mu\mathrm{m}^3\,\mathrm{s}^{-1}$ vs $Q_2 = 808\,\mu\mathrm{m}^3\,\mathrm{s}^{-1}$) despite the smaller number of the proximal flagella. The improved pumping rate in the first case is ascribed to higher capability of the proximal flagella in pumping due to their proximity to the ostium (for the case of synchronized flagella beat they contribute 64% of the total pumping, despite being only a third of

the flagella in the array). Additionally, faster proximal flagella leads to 33% less mechanical power expenditure ($\mathcal{P}_1 = 97.3\,\text{fW}$ vs $\mathcal{P}_2 = 144.9\,\text{fW}$, where $\mathcal{P}_i$ is the averaged total power expenditure by all 24 flagella over the beat cycle for the $i$-th case). Considering that pumping rate increases linearly with the beat frequency, whereas the power expenditure increases as the square of the beat frequency ($P \propto f^2$, $Q \propto f$), employing faster beating of the fewer proximal flagella can be a suitable compromise between the power expenditure and pumping rate.

The presence of both proximal and distal flagella, and their collective imparted upward momentum is essential for efficient pumping of the inflow through the collars. Without the presence of the distal flagella, the system continues to pump, but the operation of the hydrodynamic gasket is impaired and water flow through the collars is dramatically reduced. The vacant space created by the absence of distal flagella, shifts the location of the back flow from above ostium toward the vacant space. Consequently, a significant part of the inflow bypasses the collars and leaves the domain without being filtered (*Appendix 1—figure 7C*). Considering the minor effect of synchronous flagella beat in reducing the power expenditure, the required imparted momentum of the distal flagella can potentially also be achieved with flagella having a different waveform, for example different wavelength, as long as the averaged upward flow velocity due the presence of the flagella is able to maintain the hydrodynamic gasket in place (*Appendix 1—figure 7D*). We find that the volume flow rate is affected by the wavelength, with an optimum wavelength of 5 µm (*Appendix 1—figure 8*).

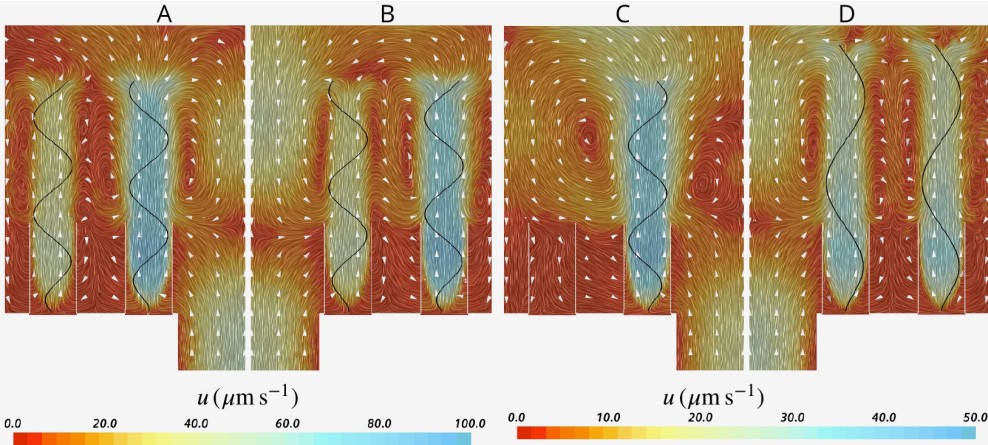

**Appendix 1—figure 7.** Hydrodynamic interaction between the flagella. **A** And **B** show average velocity fields for the cases with doubled beat frequency on the proximal compared to the distal flagella ($f_p = 60$ and $f_d = 30\,\text{Hz}$), and vice versa (**A** and **B**, respectively). Asynchronization between the proximal and distal flagella does not impair pumping. It does, however, affect the pumping rate, but the effect is due to the unequal contribution of the proximal and distal flagella on the pumping. (**C**) The presence of the distal flagella and thus their imparted momentum is important for creating the recirculation in the core of the region, without which the location of recirculation is shifted. The momentum may be imparted by flagella with different wavelength. (**D**) Averaged velocity field for wavelength $\lambda = 9\,\mu\text{m}$.

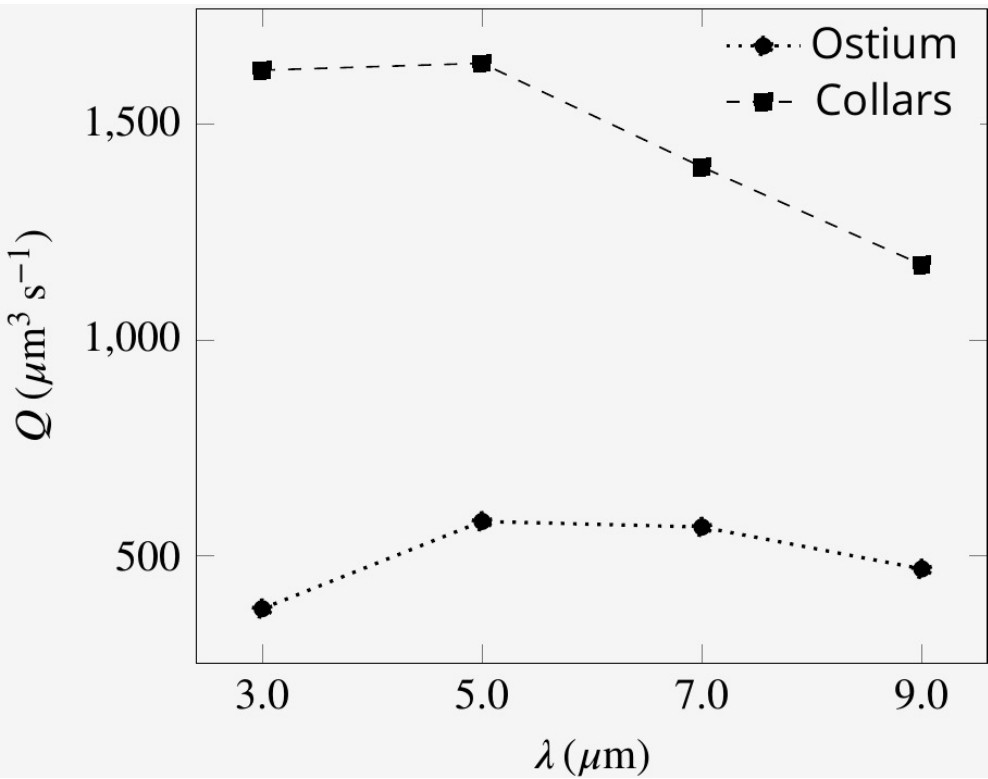

**Appendix 1—figure 8.** Pumping rate and volume flow rate through the collars for flagella with different wavelength.

## Appendix 2

### Inclusion of gasket into the model

We model the gasket as an impermeable plate that forms a canopy at the tip of the collars as depicted in *Appendix 2—figure 1*.

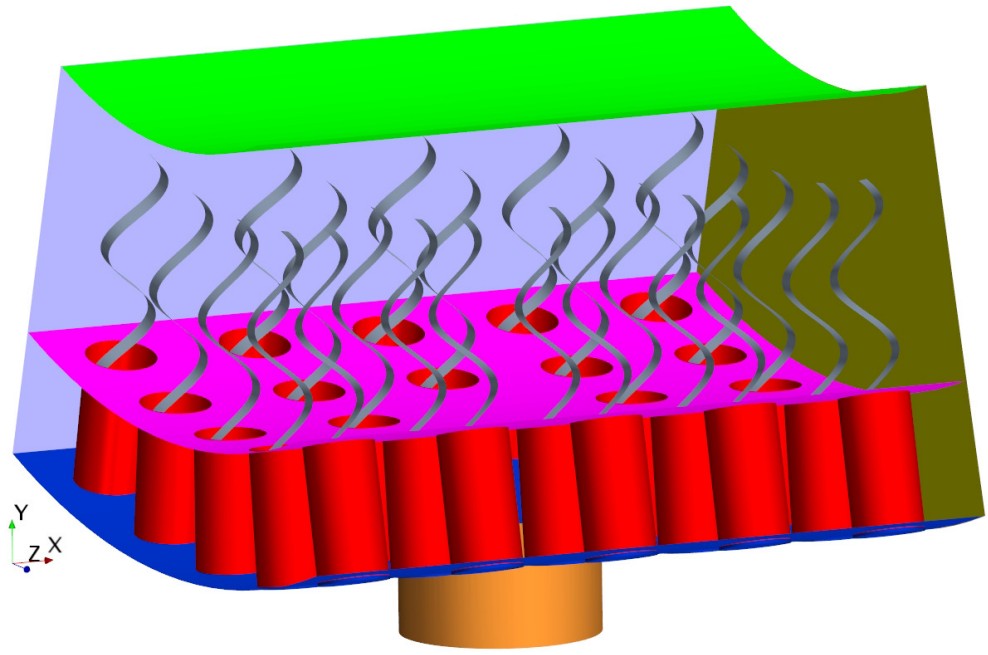

**Appendix 2—figure 1.** The simulation unit with presence of the gasket (pink).

### Moving computational mesh

We use mesh morphing along with the overset method to move the computational mesh corresponding to the motion of the flagella. With the overset method, rather than moving the entire mesh, it deforms the mesh only around the flagella, so-called the overset region, which significantly reduces the computational cost. A background stationary mesh including the collars is also generated which is superimposed by the overlapping overset mesh. The two mesh regions are implicitly coupled, the field data are interpolated back and forth between the two meshes (i.e. the overset and background meshes) to generate a smooth solution at each iteration.

### Visualization of the flow field in the inner core of the chamber

Once fluid leaves the computational domain vertically into the core of choanocyte chamber, it is accumulated and directed axially toward one end of the chamber (apopyle) where it is open. To illustrate this, we model the flow in the cylindrical core of the chamber *Figure 2A* in the main text. The flow enters the cylinder from the lateral surface and one end (velocity inlet boundary of 1 $\mu ms^{-1}$, in the range of the mean velocity of flow leaving the wedge geometry of *Figure 2*), and exit from the other end (uniform pressure boundary). *Figure 2* shows the velocity and pressure fields in a cross-section oriented axially in the center of the domain. Lateral flow and shear close to sides (hence right above flagella tip) is relatively weak, and we assume that the flagella action is not strongly affected.

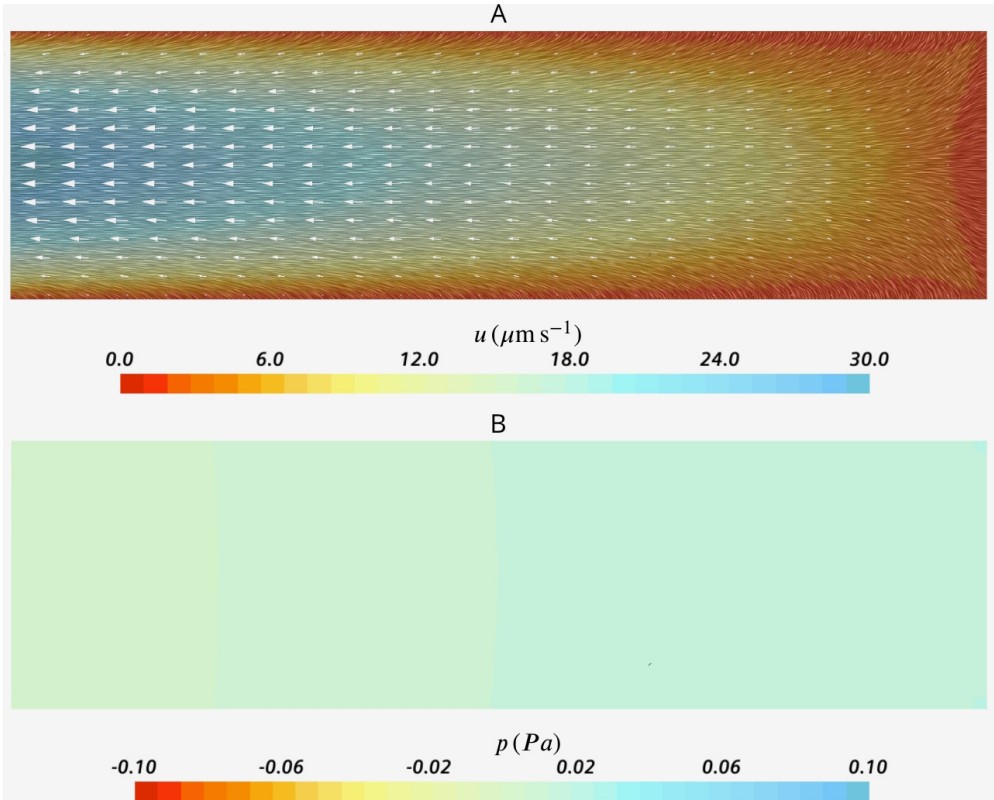

**Appendix 2—figure 2.** Velocity (A) and pressure (B) fields in the cross section of the inner core cylinder. The modeled cylinder is 200 µm long (=8 wedge-unit long), with diameter $D_{core} = D_{chamber} - 2 * h_{wedge} = 55\,\mu\mathrm{m}$, where $D_{chamber}$ is the diameter of chamber and $h_{wedge}$ the height of the wedge geometry of **Figure 2**.

## Independence of solution from the global boundary conditions

*Appendix 2—figure 4A* shows snapshots of the velocity field in three middle units (separated by the white dashed lines) of the domain. The velocity field is virtually periodic in all these units indicating the minor effect of the domain ends on each single units, independent of their location with respect to the ends. *Appendix 2—figure 4B* shows the snapshots of the pressure field in the middle unit. Despite high variation inside the ostium and between the flagella, the pressure is nearly uniform above the flagella.

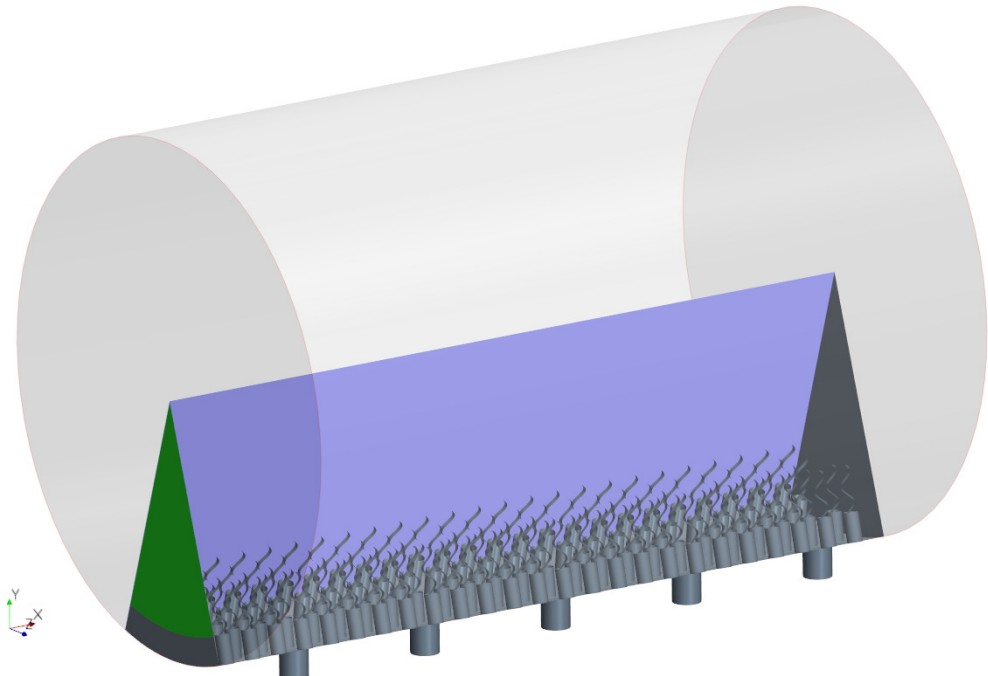

**Appendix 2—figure 3.** The simulation domain in an extended slice of the cylindrical chamber including five pumping units, that is, 5 ostia and 120 collar-flagella, closed (no-slip BC) at one end but open (uniform pressure BC) at the other end (green).

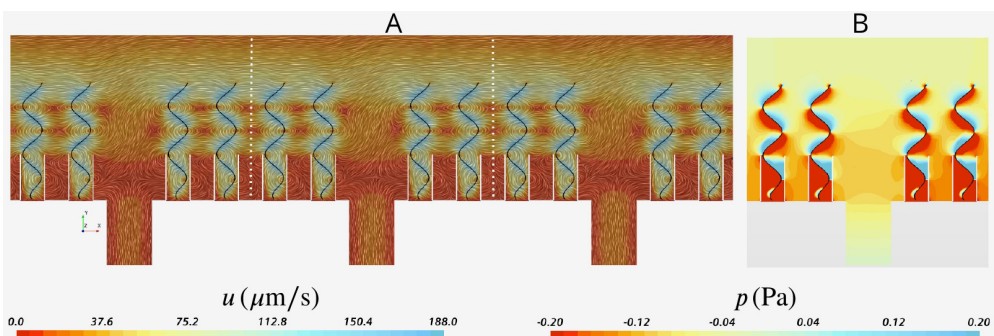

**Appendix 2—figure 4.** Snapshot of velocity and pressure fields for the simulation domain including five units depicted in *Appendix 2—figure 3B*. (**A**) Velocity field in the three middle units (separated by white dash lines) shows the periodicity of the flow fields in these units independent of their location with respect to the outlet to the left (apopyle). (**B**) Pressure field in one single unit illustrating uniformity of the pressure above the flagella. Accordingly, the computational domain is reduced to one single unit using periodic and uniform pressure boundary conditions as shown in *Figure 2*.

## Mesh convergence study

In order for the solution to be independent of the mesh sizes, the CFD simulations are conducted at different mesh size. Because the problem is quasi-steady, the mesh convergence study has been performed at one arbitrary time step. *Appendix 2—figure 5* shows the independence of the solution from the mesh sizes for the base case.

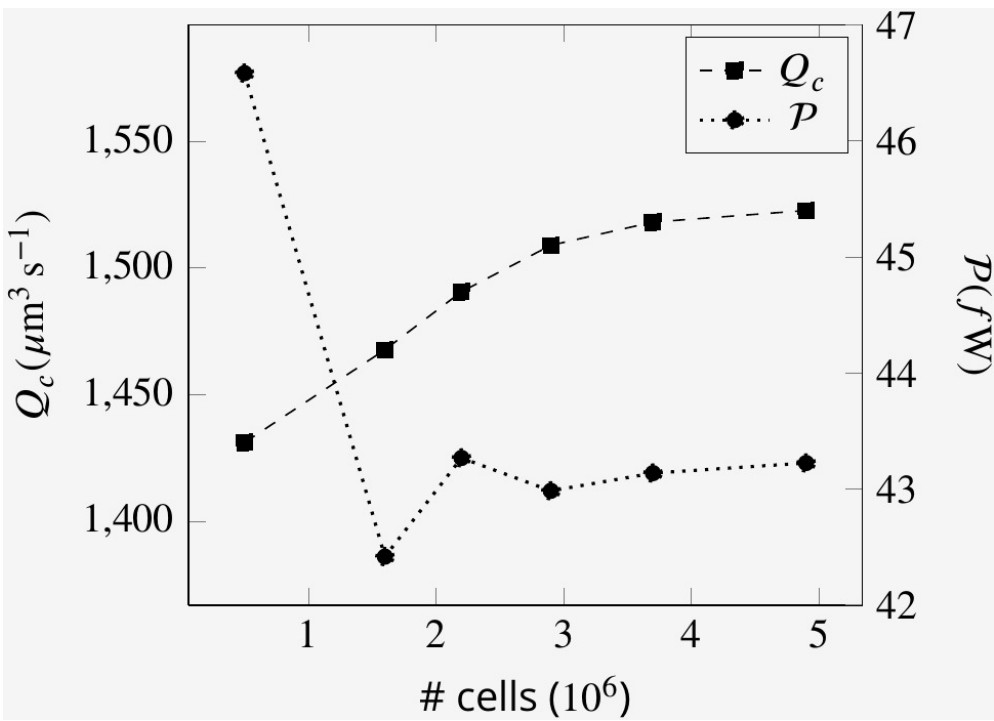

**Appendix 2—figure 5.** Mesh size independence of volume flow rate through the collars $Q_c$ and the power expenditure by the flagella $\mathcal{P}$ at an arbitrary time step.

