## [Decision Letter]

**Acceptance summary:**

The fluid dynamics of filter feeding in sponges is a complex problem on many length scales, from the flagella of filter-feeding cells within small chambers to the large scale flow past the entire sponge. Asadzadeh et al. address a significant puzzle in the physiology of sponges lacking the "gasket" which, in other species, diverts flow toward the filtration devices on the feeding cells. Through a combination of experimental work and numerical computations the authors demonstrate the existence of an unusual flow pattern that accomplishes this diversion, and suggest implications for our understanding of the evolution of the first metazoans.

**Decision letter after peer review:**

Thank you for submitting your article "Hydodynamics of sponge pumps and evolution of the sponge body plan" for consideration by *eLife*. Your article has been reviewed by three peer reviewers, one of whom is a member of our Board of Reviewing Editors, and the evaluation has been overseen by Aleksandra Walczak as the Senior Editor. The reviewers have opted to remain anonymous.

The reviewers have discussed the reviews with one another and the Reviewing Editor has drafted this decision to help you prepare a revised submission.

As the editors have judged that your manuscript is of interest, but as described below that substantial revisions are required before it is published, we would like to draw your attention to changes in our revision policy that we have made in response to COVID-19 (https://elifesciences.org/articles/57162). First, because many researchers have temporarily lost access to the labs, we will give authors as much time as they need to submit revised manuscripts. We are also offering, if you choose, to post the manuscript to bioRxiv (if it is not already there) along with this decision letter and a formal designation that the manuscript is "in revision at *eLife*". Please let us know if you would like to pursue this option. (If your work is more suitable for medRxiv, you will need to post the preprint yourself, as the mechanisms for us to do so are still in development.)

Summary:

The authors present a numerical and experimental investigation of the pumping mechanism in filter feeding sponges. The feeding is performed by choanocytes, which comprise a pumping flagellum and a feeding collar that protrude from the wall of the feeding chambers, which has periodically arranged holes through which unfiltered fluid can enter when pulled by the flagellar action.

In this study, the mechanism is examined for the case of sponges that do not possess a "gasket" (canopy-structure) over these feeding collars. The key question that the authors examine is, then, "how is the flow forced through the collars (allowing particle capture), when there is no gasket to stop it simply going straight to the centre of the chamber"?

The answer given is that just as flow is pulled through the holes, a reverse flow (which arises from the condition that flow in the chamber is, at least locally, divergence free), forms above the hole, which creates a stagnation point that pushes the incoming fluid laterally out through the feeding collars.

Essential revisions:

1) On the whole we found this to be a solid paper that addresses an important question by means of computational studies. Having said that, we are disappointed at the lack of physical interpretation given to the setup of the system and the results. In essence, the papers reads like an experimental study of the system with essentially no interpretation of the magnitudes of any of the physical quantities calculated, particularly in light of the fact that sponge hydrodynamics is not a commonly studied subject for the readers of *eLife*. The authors need to do a serious job of re-presentation of the results to establish the physical significance of various pressures (are these arising from Bernoulli-like pressure differences across the sponge body due to the external flow?), flow rates (say, by comparing them to Poiseuille flow), powers (comparing to the flagellar input power), etc. There is very little in the way of dimensional analysis to make the results intuitively clear. As such, the very many numerical values of physical quantities have no context.

2) From a fluid dynamics perspective, the fundamental result is fairly easy to see, and is simply a manifestation of local mass conservation (though of course retrodiction is of course much easier than prediction). The presentation would be greatly aided by a very simple schematic of the streamlines from two forces above a wall, either side of a hole, below a no-penetration (not no slip) boundary. We would have liked to have seen a reduced model of this kind involving stokeslets over a wall with a gap, which the authors should consider looking at, but this is not is a necessity for publication.

3) The description of the "flagellar vane" is a bit arcane. We are familiar with standard flagellae, and flagellae with mastigonemes, but have not come across this sort of ultrastructure before (is this a uniquely choanoflagellate/sponge structure). Reading other papers helped, but the manuscript needs a clearer description of these 3 types of flagellae, particularly when the image in Appendix 1—figure 1C seems to have ordinary flagellae.

4) An image of gasket-type vs. non gasket type would have helped our understanding of the motivation for the study earlier on. In general, we found that it was only possible to understand what was actually going on by the end of the paper, so it would be good to go through the Introduction once more to make sure it is as clear as possible to the more general reader.

5) The flagellum model (Equation 1) is not arclength preserving. Reading a previous paper it seems that the total change in arclength is only a few percent over a beat, however the velocities on the flagellum will also be subtly wrong, as material points will not be exhibiting the characteristic figure of eight motion. We have convinced ourselves that this doesn't affect the results too significantly (any error would be smaller than the error associated with taking very difficult experimental measurements), so we will not insist on a rerunning of the results, but it would not have been hard to make the waveform arclength preserving.

6) We are confused by the wedge geometry of Figure 2. If fluid leaves the computational domain "vertically" then how is there any net transport of fluid along the long axis of the ascon? Would not the flagella be strongly perturbed by the lateral flow?

7) No detail is given at all as to how the flagella beating is modelled in the CFD code. Are there regularized Stokeslets? Is slender body hydrodynamics used? Please clarify.

8) We strongly suggest the authors avoid the rainbow colourmap wherever possible, as it is not perceptually uniform (see https://peterkovesi.com/projects/colourmaps/) and is also generally bad for people who are colour blind.

9) The authors end up documenting a large number of individual differences between ascon and leucon sponges, but only study them in isolation – and only a subset of them. Compared to leucons, ascons have: (1) no physical gasket (2) shorter collars and (3) one large incurrent pore for 24 choanocytes (per choanocyte chamber), instead of many small pores (one per choanocyte.) Only the third difference is usually considered part of the definition of leuconoid vs. asconoid type, but the paper focuses almost entirely on the first and second differences. We recommend that the authors comment on the third parameter as well – if only by drawing a comparison with their earlier, published model.

10) Regarding the total difference of overall efficiency between the leuconoid and asconoid organizations, the discussion is a bit unclear. The authors state that "Our results indicate that reduced resistance in the ascon and sycon-type aquiferous systems comes with reduced volume filtered for high retention efficiency." But the paper performs no explicit comparison of filtered volumes between sycon and leucons until the very end, and even then it concludes that these are actually not really different: "Furthermore, both experimental and CFD estimates of flow rate per choanocyte [in sycons] are comparable to published estimates for leucon sponges that range from 17 to 236 *µ*m3 s^-1^ for different species of demosponges and glass sponges (Larsen and Riisgård, 1994; Leys et al., 2011; Ludeman et al., 2017), suggesting similar pumping capacity despite different pumping mechanism." This contradicts the statement of the Introduction – even though the highest pumping capacity reported for a leuconoid sponge (236 *µ*m3 s^-1^ per choanocyte) is higher than the one reported for syconoid sponges (48 *µ*m3 s^-1^ per choanocyte) suggesting that the leucon organization has at least higher potential.

We think the authors should settle on one of their two interpretations, or alternatively state that both are valid alternatives. In particular, if leucons do not have higher filtration capacity, what advantage do they have over ascons and sycons (which additionally do not risk clogging, as the authors mention)? What is the specific niche of leuconoid sponges?

11) Figure 2 shows that all flagella within the chamber beat in the same plane (with all vanes parallel to each other), parallel to the long axis. Are there reasons to think this reflects biological reality or is it a simplification of the model?

12) Subsection “Pumping mechanism”: the vane width considered is in the 0.3-0.7 *µ*m range. How was this range of values decided? Is it close to real vanes of calcareous sponges? Why does it stay so far below the collar diameter (2.5 *µ*m)?

13) Some calcareous sponges have a leuconoid organization, yet lack a physical gasket. Can the authors comment on how they think these sponges work?

---

## [Author Response]

Essential revisions:1) On the whole we found this to be a solid paper that addresses an important question by means of computational studies. Having said that, we are disappointed at the lack of physical interpretation given to the setup of the system and the results. In essence, the papers reads like an experimental study of the system with essentially no interpretation of the magnitudes of any of the physical quantities calculated, particularly in light of the fact that sponge hydrodynamics is not a commonly studied subject for the readers of eLife. The authors need to do a serious job of re-presentation of the results to establish the physical significance of various pressures (are these arising from Bernoulli-like pressure differences across the sponge body due to the external flow?), flow rates (say, by comparing them to Poiseuille flow), powers (comparing to the flagellar input power), etc. There is very little in the way of dimensional analysis to make the results intuitively clear. As such, the very many numerical values of physical quantities have no context.

We agree that there is a lack of physical interpretation of the system and the results. A major obstacle in the way of such interpretation is the fact that, the flow regime is highly viscous (low Reynolds number regime), and most of the energy is dissipated inside the chamber. For example, in the current case, less than 1% of the flagella power input is converted to pressure delivered to overcome the flow through the tubular ostium, the rest is dissipated due to large velocity gradients associated with recirculation and pressure loss through collars. As a partial remedy, we considered the pump curve characteristics and provide a plot of pressure vs. flow rate for the pumping unit and the canal system.

Beside revealing and highlighting the hydrodynamic gasket that ensures an efficient prey retention, another goal of the current manuscript is to investigate (by employing typical dimensions of a model organism) whether ascons and sycons would be able to pump and filter enough water, if they truly lack a physical gasket, and if so, what are the determining factors and trade-offs.

An entire new subsection (“Pump curve characteristic”) has been written, in an attempt to give physical interpretation of the pumping system both in ascons/sycons, and leucons.

2) From a fluid dynamics perspective, the fundamental result is fairly easy to see, and is simply a manifestation of local mass conservation (though of course retrodiction is of course much easier than prediction). The presentation would be greatly aided by a very simple schematic of the streamlines from two forces above a wall, either side of a hole, below a no-penetration (not no slip) boundary. We would have liked to have seen a reduced model of this kind involving stokeslets over a wall with a gap, which the authors should consider looking at, but this is not is a necessity for publication.

We agree with reviewers that simple models would greatly aid the presentation, and we tried our best to make the CFD model as simple as possible, to the point that the key flow structures, both qualitatively and quantitatively, are preserved. However, as now explained in the new subsection “Pump characteristic curve”, we believe the CFD unit in Figure 2 is an integrated basic pumping unit of the asconoid and syconoid sponges, and further simplification would not explain the phenomena. For example, as suggested, we considered and modeled two stokeslets above a wall (at the height of *h_s_*= 4.7*µ*m, corresponding to the length of collar filters), either side of a hole (at horizontal distances of *x_i_*= ±*iµ*m), below a no-penetration boundary (at the height of *h_np_*= 42.5*µ*m, corresponding to the radius of the choanocyte chamber). Author response image 1 shows the streamlines for two cases (*x*_5_ and *x*_10_, corresponding to the horizontal location of flagella with respect to the hole). In the first case (Author response image 1), there will be no recirculation created above the hole. A weak circulation region is formed above the hole once the stokeslets are positioned far away from the hole (case *x*_10_, Author response image 1); this recirculation shows elements of the hydrodynamic gasket: with downflow to a stagnation point (about 1-2*µ*m above the inlet) and radially directed outflow (the remaining flow may be ignored). It is noted that the pumping through the hole drops dramatically (Author response image 1) as the stokeslets are positioned away from the hole.

The real problem is more complex due to many forces distributed along the beating flagella (Figure 2); As shown, e.g., in Appendix 1—figure 7C, the presence of both proximal and distal flagella, and their collective imparted upward momentum is essential for efficient pumping of the inflow through the collars. Additionally, the flow field is markedly dependent on the presence of porous structure (collar filter) around force generating flagella (e.g. Figure 4).

**Author response image 1. respfig1:** Streamlines and pumping rate of two stokeslet above a wall (at the height of *h_s_*= 4.7*µ*m), either side of a hole, below a no-penetration boundary. Note that due to the symmetry, the results are shown only in half of the plane. A weak recirculation region is formed above the hole as the horizontal distances *x_i_*increases (*x*_5_ = ±5*µ*m (A) to *x*_10_ = ±10*µ*m (B)), but it’s formation comes at the cost of a dramatically reduced pumping rate into the hole (C).

3) The description of the "flagellar vane" is a bit arcane. We are familiar with standard flagellae, and flagellae with mastigonemes, but have not come across this sort of ultrastructure before (is this a uniquely choanoflagellate/sponge structure). Reading other papers helped, but the manuscript needs a clearer description of these 3 types of flagellae, particularly when the image in Appendix 1—figure 1C seems to have ordinary flagellae.

Flagella vanes are believed to be present in all sponges, and have been well illustrated in both demosponges and Hexactinellids.

A new paragraph has been added in the Introduction clarifying the differences between mastigonemes and vanes. Also a recently taken image is now been added in the appendix (Appendix 1—figure 2) visualizing the vane in calcareous sponge *S. coactum* for the first time.

4) An image of gasket-type vs. non gasket type would have helped our understanding of the motivation for the study earlier on. In general, we found that it was only possible to understand what was actually going on by the end of the paper, so it would be good to go through the Introduction once more to make sure it is as clear as possible to the more general reader.

We agree and now explain the concept of gasket in the Introduction.

5) The flagellum model (Equation 1) is not arclength preserving. Reading a previous paper it seems that the total change in arclength is only a few percent over a beat, however the velocities on the flagellum will also be subtly wrong, as material points will not be exhibiting the characteristic figure of eight motion.We have convinced ourselves that this doesn't affect the results too significantly (any error would be smaller than the error associated with taking very difficult experimental measurements), so we will not insist on a rerunning of the results, but it would not have been hard to make the waveform arclength preserving.

We agree, indeed, in our model, the flagellum would be extensible and its length varies slightly during the beat cycle (ca. 2%), resulting in marginally wrong velocities on the flagellum. However, implementing Equation 1 in the numerical scheme is computationally cheaper than preserving the arclength. This is because the latter requires importing a table of motion data and interpolating the velocities of each point on the flagellum at each time step. To exemplify the insignificant error involved, we reran the simulation results for the base case with an inextensible flagellum that preserves the arc length. Appendix 1—figure 3 shows two snapshots of velocity field for an extensible vs. inextensible flagellum. Although the velocity of the flagellum slightly differ, the flow field in the domain is insignificantly affected with less than 4% difference in the pumping rate.

It has also been argued [3] that the extensible and inextensible sheets have the same swimming velocity (hence the same generated force) to leading order in *ak*, where *a* is the amplitude of the beat, and *k* the wavenumber.

6) We are confused by the wedge geometry of Figure 2. If fluid leaves the computational domain "vertically" then how is there any net transport of fluid along the long axis of the ascon? Would not the flagella be strongly perturbed by the lateral flow?

Once fluid leaves the computational domain vertically into the core of the choanocyte chamber, it is directed axially toward one open end of the chamber (apopyle). To illustrate this, we model the flow in the cylindrical core of the chamber, closed at the right end and open through the apopyle at the other end. The flow enters the cylinder from the lateral surface and one end (velocity inlet boundary of 1*µ*ms^−1^, which is in the range of the mean velocity of flow leaving the wedge geometry of Figure 2), and exit from the other end (uniform pressure boundary). Appendix 2—figure 2 shows the velocity and pressure fields in a cross section oriented axially in the center of the domain. Lateral flow and shear close to sides (hence right above flagella tip) is relatively weak, and we assume that the flagella action is not strongly affected.

Furthermore, considering an extended computational domain including 5 pumping units (Appendix 2—figure 3), we illustrate that the boundary conditions applied on the wedge geometry is independent of the global boundary conditions of the entire chamber, hence from the directional axial flow in the core region (Appendix 2—figure 4 and Appendix 2—figure 2B).

Appendix 2—figure 2 is referred to in the subsection “Morphology of the flagellated chamber”.

7) No detail is given at all as to how the flagella beating is modelled in the CFD code. Are there regularized Stokeslets? Is slender body hydrodynamics used? Please clarify.

The numerical method is the finite volume discretization and solution of the Navier-Stokes equations does not involve singularities, as described in the Materials and methods. But we agree that more clarification is needed, specifically on how the flagella and corresponding computational cell motion is modeled.

More explanation has been added in the subsection “Morphology of the flagellated chamber” and in the Materials and methods.

8) We strongly suggest the authors avoid the rainbow colourmap wherever possible, as it is not perceptually uniform (see https://peterkovesi.com/projects/colourmaps/) and is also generally bad for people who are colour blind.

We thank the reviewers for this important comment. We have changed all the colour maps from the rainbow to the Kelvin temperature.

9) The authors end up documenting a large number of individual differences between ascon and leucon sponges, but only study them in isolation – and only a subset of them. Compared to leucons, ascons have: (1) no physical gasket (2) shorter collars and (3) one large incurrent pore for 24 choanocytes (per choanocyte chamber), instead of many small pores (one per choanocyte.) Only the third difference is usually considered part of the definition of leuconoid vs. asconoid type, but the paper focuses almost entirely on the first and second differences. We recommend that the authors comment on the third parameter as well – if only by drawing a comparison with their earlier, published model.

We agree with the reviewers that more clarification is needed regarding the difference between the two basic pumping units. To the best of our knowledge, even in leucons there are several (less than 10) choancytes per incurrent pore. The difference, though, is that, in leucons, the basic pumping units differ from those of ascons or sycons. The basic pumping units, are the simplest subdivision of choanocyte chamber exposed to the same pressure difference resulting from the canal system. Hence, these units all work in parallel. For sponges lacking a physical gasket, the unit is one ostium with several neighbouring choanocytes (Figure 2B). Presence of a physical gasket (leucons), however, leads to two zones of high and low pressure inside the chambers [5, 2], effectively exposing each individual choanocyte to the same pressure difference. Therefore, in leucons, the basic pumping unit is a further reduced subdivision of chamber including only one choanocyte, irrespective of the location of the choanocyte with respect to the ostium [2].

These differences are now clarified and highlighted in the new added subsection “Pump characteristic curve”.

10) Regarding the total difference of overall efficiency between the leuconoid and asconoid organizations, the discussion is a bit unclear. The authors state that "Our results indicate that reduced resistance in the ascon and sycon-type aquiferous systems comes with reduced volume filtered for high retention efficiency." But the paper performs no explicit comparison of filtered volumes between sycon and leucons until the very end, and even then it concludes that these are actually not really different: "Furthermore, both experimental and CFD estimates of flow rate per choanocyte [in sycons] are comparable to published estimates for leucon sponges that range from 17 to 236 µm3 s^-1^ for different species of demosponges and glass sponges (Larsen and Riisgård, 1994; Leys et al., 2011; Ludeman et al., 2017), suggesting similar pumping capacity despite different pumping mechanism." This contradicts the statement of the Introduction – even though the highest pumping capacity reported for a leuconoid sponge (236 µm3 s^-1^ per choanocyte) is higher than the one reported for syconoid sponges (48 µm3 s^-1^ per choanocyte) suggesting that the leucon organization has at least higher potential.We think the authors should settle on one of their two interpretations, or alternatively state that both are valid alternatives. In particular, if leucons do not have higher filtration capacity, what advantage do they have over ascons and sycons (which additionally do not risk clogging, as the authors mention)? What is the specific niche of leuconoid sponges?

The latter statement referred to the pumping capacity of the choanocyte, but the first statement in the Introduction, to the pumping capacity of the whole sponge in ascons and sycons, which has fewer choanocytes (compared to leucons) due to the architecture of the chambers. Still, we agree that our comments on the filtration capacity do not seem coherent. The statement in the Introduction was removed to avoid any inconsistency in the interpretation.

The architecture of leuconid sponges allows the sponge to grow larger and have greater surface area. However, we think that the large surface area does take a lot of energy to maintain and so generally leucons thrive in food rich environments.

11) Figure 2 shows that all flagella within the chamber beat in the same plane (with all vanes parallel to each other), parallel to the long axis. Are there reasons to think this reflects biological reality or is it a simplification of the model?

Indeed, this is a simplification of the model. In the manuscript, we have illustrated that asynchronization among flagella does not matter, but this was done only for cases where all flagella beat in the same plane. Following the reviewers comments, we have also examined whether the pumping and average flow structure would be significantly affected, even if some flagella beat in a different plane. We considered the same computational domain as in Figure 2, where now every 2nd column of flagella beat in a perpendicular plane (Author response image 2). The results show less than 3% difference in the pumping rate into the chamber as a result of not all flagella beating in the same plane.

**Author response image 2. respfig2:** Computational geometry where some flagella beat in a perpendicular plane.

The results for the new simulation case have been added to the subsection “Asynchronization among flagella”, and referred to in the main text.

12) Subsection “Pumping mechanism”: the vane width considered is in the 0.3-0.7 µm range. How was this range of values decided? Is it close to real vanes of calcareous sponges? Why does it stay so far below the collar diameter (2.5 µm)?

A base width of 0.7 *µ*m has been chosen based on an image showing the remnant of vane in calcareous ascon *Leucosolenia eleanor* (Author response image 3). However, to the best of our knowledge, at the time of conducting the simulations, there was no data on the actual dimension of vane in calcareous sponges, despite their presence being reported [1, 4]. We initially present the result for a range of different vane width 0.3-0.7 *µ*m. However, we also consider cases where the vane width is 1.4 *µ*m on the unconfined part of flagellum, but 0.7 *µ*m on the confined part. Width of 0.7 *µ*m is technically the maximum width (on the confined part of the flagellum) which allows the flagellum motion without becoming in contact with the collar. Although some leucons have vanes that span the collar which leads to a high pressure ’positive displacement pump’ thanks to their physical gasket, vanes spanning the collar in sycons would not be important because of the different pumping mechanism (Figure 4).

Furthermore, a recently taken image has now been added as Appendix 1—figure 2 visualizing the vane in *S. coactum* indicating a vane width of ∼ 1*µ*m.

**Author response image 3. respfig3:** SEM image of calcareous ascon *Leucosolenia eleanor* showing flagellum with remnants of about ∼ 0.7*µ*m wide vanes extending over the full flagellum length.

The range has been increased to 0.3-1.4 *µ*m, and Figure 3 has been updated, and as mentioned above, a new image visualising the vane has been added in Appendix 1.

13) Some calcareous sponges have a leuconoid organization, yet lack a physical gasket. Can the authors comment on how they think these sponges work?

Although sponges with a complex canal system appear to have a ’gasket-present’ pumping unit, a ’gasket-absent’ unit can still function efficiently if connected to a complex, yet open and less resistive canal system, or inefficiently if connected to an open canal system, but with highly resistive ostia. Therefore, there could well be sponges characterized as leuconoid types yet missing a physical gasket, or as syconoid types but with a physical gasket. The appearance, structure and nature of the gasket would also depend on the hydrodynamics dictated by the canal system. Therefore, further research, using better preservation and imaging techniques, is required to elucidate the exact nature of the sealant element, and to investigate the role that hydrodynamics is playing in forming the seal in different sponges. Recent observations of *sycon coactum* (Appendix 1—figure 2) have shown some mesh structures that could have a similar (gasket) effect of guiding flow to filters.

The above explanation has been added at the end of the subsection “Pump curve characteristic”.

**References:**

1) Bjorn A. Afzelius. Flimmer-flagellum of the sponge. Nature, 191(4795):1318, 1961.

2) Sayed Saeed Asadzadeh, Poul S. Larsen, Hans Ulrik Riisgard, and Jens H. Walther. Hydrodynamics of the leucon sponge pump. J. Roy. Soc. Inter, 16(150):20180630, 2019.

3) Eric Lauga and Thomas R. Powers. The hydrodynamics of swimming microorganisms. Rep. Prog. Phys., 72(9):096601, 2009.

4) TL Simpson. The cell biology of sponges. Springer, Berlin Heidelberg, New York, 1984.

5) Norbert Weissenfels. The filtration apparatus for food collection in freshwater sponges (Porifera, Spongillidae). Zoomorphology, 112(1):51–55, 1992.